# AB-UPT: Scaling Neural CFD Surrogates for High-Fidelity Automotive Aerodynamics Simulations via Anchored-Branched Universal Physics Transformers

**Benedikt Alkin**[*,1]**, Maurits Bleeker**[*,1]**, Richard Kurle**[*,1]**, Tobias Kronlachner**[*,1]**,**
**Reinhard Sonnleitner**[1]**, Matthias Dorfer**[1]**, Johannes Brandstetter**[1,2]

[*]Equal contribution   [1]Emmi AI   [2]ELLIS Unit, LIT AI Lab, JKU Linz
Correspondence to `johannes@emmi.ai`

Reviewed on OpenReview: `https://openreview.net/forum?id=nwQ8nitlTZ`

## Abstract

Recent advances in neural surrogate modeling offer the potential for transformative innovations in applications such as automotive aerodynamics. Yet, industrial-scale problems often involve volumetric meshes with cell counts reaching 100 million, presenting major scalability challenges. Complex geometries further complicate modeling through intricate surface-volume interactions, while quantities such as vorticity are highly nonlinear and must satisfy strict divergence-free constraints. To address these requirements, we introduce Anchored-Branched Universal Physics Transformers (AB-UPT) as a novel modeling scheme for building neural surrogates for computational fluid dynamics (CFD) simulations. AB-UPT is designed to: (i) decouple geometry encoding and prediction tasks via multi-branch operators; (ii) enable scalability to high-resolution outputs via neural simulation in a low-dimensional latent space, coupled with *anchored* neural field decoders to predict high-fidelity outputs; (iii) enforce physics consistency by a divergence-free formulation. We show that AB-UPT yields state-of-the-art predictive accuracy of surface and volume fields on automotive CFD simulations ranging from 33 thousand up to 150 million mesh cells. Furthermore, our anchored neural field architecture enables the enforcement of hard physical constraints on the physics predictions without degradation in performance, exemplified by modeling divergence-free vorticity fields. Notably, the proposed models can be trained on a single GPU in less than a day and predict industry-standard surface and volume fields within seconds. Additionally, we show that the flexible design of our method enables neural simulation from a computer-aided design geometry alone, thereby eliminating the need for costly CFD meshing procedures for inference.

## 1 Introduction

Computational fluid dynamics (CFD) is central to automotive aerodynamics, offering in-depth analysis of entire flow fields and complementing wind tunnels by simulating open-road conditions. The fundamental basis of almost all CFD simulations are the Navier-Stokes (NS) equations, describing the motion of viscous fluid substances around objects. However, the computational cost of solving the NS equations necessitates modeling approximations, most notably regarding the onset and effects of turbulence. Therefore, CFD employs different turbulence modeling strategies, balancing accuracy and cost. In this context, seminal datasets, such as DrivAerNet (Elrefaie et al., 2024a;b) and DrivAerML (Ashton et al., 2024b) have been released, allowing for an in-depth study of deep learning surrogates for automotive aerodynamics. Especially, DrivAerML runs high-fidelity CFD simulations on 140 million volumetric cells with Hybrid RANS-LES (HRLES) (Spalart et al., 2006; Chaouat, 2017; Heinz, 2020; Ashton et al., 2022), which is the highest-fidelity CFD approach routinely deployed by the automotive industry (Hupertz et al., 2022; Ashton et al., 2024b).

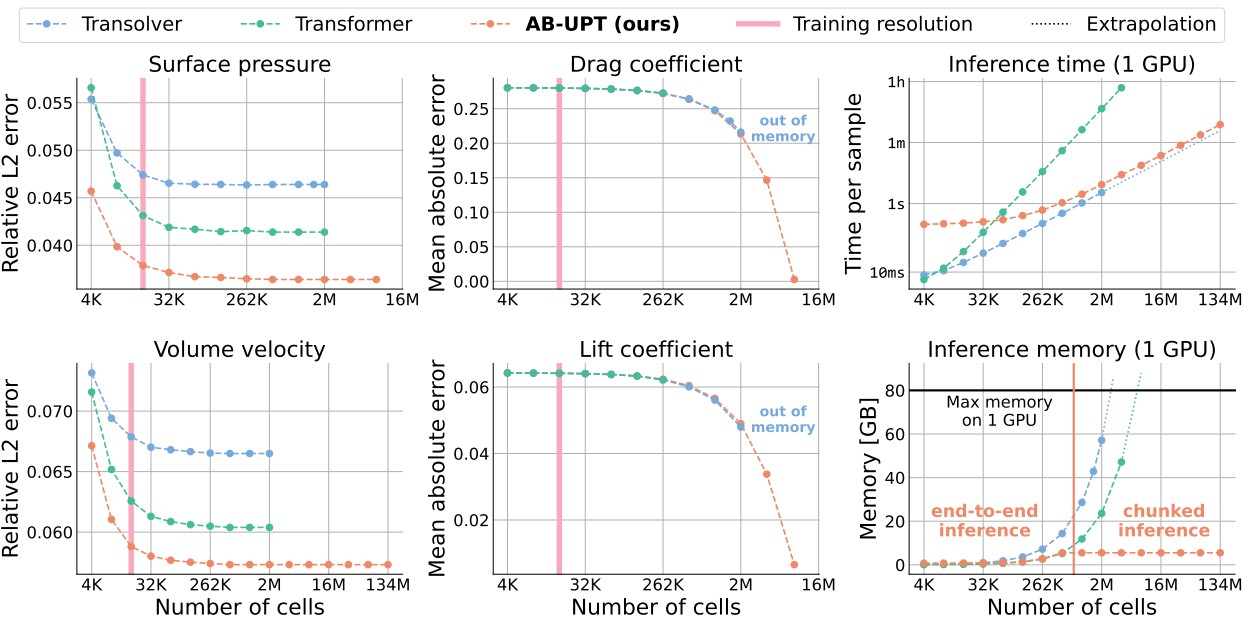

Figure 1: AB-UPT is a neural surrogate model trained to jointly model surface and volume variables of automotive CFD simulations with > 100M simulation mesh cells. It obtains state-of-the-art surface and volume predictions (left), accurately models drag and lift coefficients (center), all on a single GPU (right).

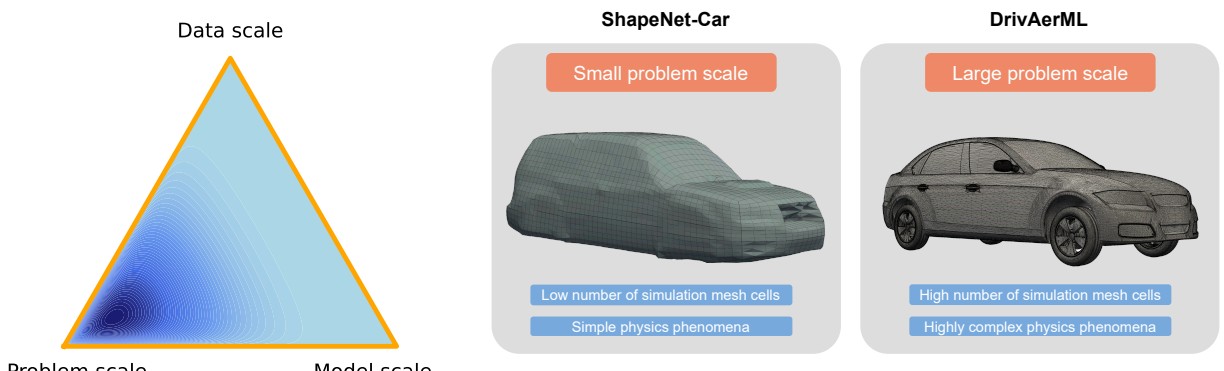

(a) **Triad of scales:** illustration of key scales in CFD surrogate modeling, placing special emphasis on the problem scale.

(b) **Problem scale** entails the fidelity of the numerical simulation and the complexity of the physics phenomena. Industry-standard simulations often use hundreds of millions of mesh cells to capture complex physics.

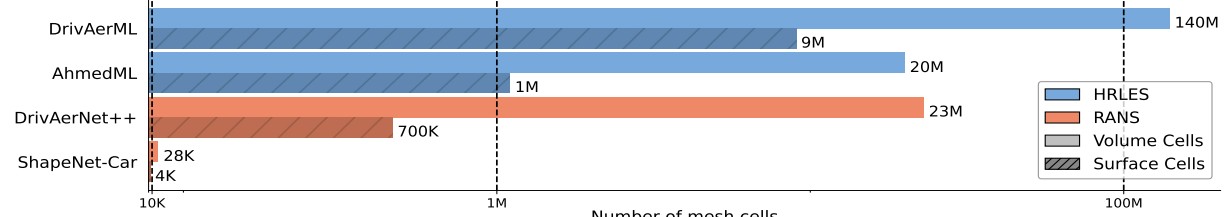

(c) The scale of CFD datasets in automotive aerodynamics, by the number of mesh cells.

Figure 2: We introduce *problem scale* in the context of computational fluid dynamics (CFD) simulations and develop a neural surrogate architecture that can handle data from publicly available CFD datasets that uses a high-fidelity CFD approach (HRLES) routinely employed in industry-standard automotive simulations.

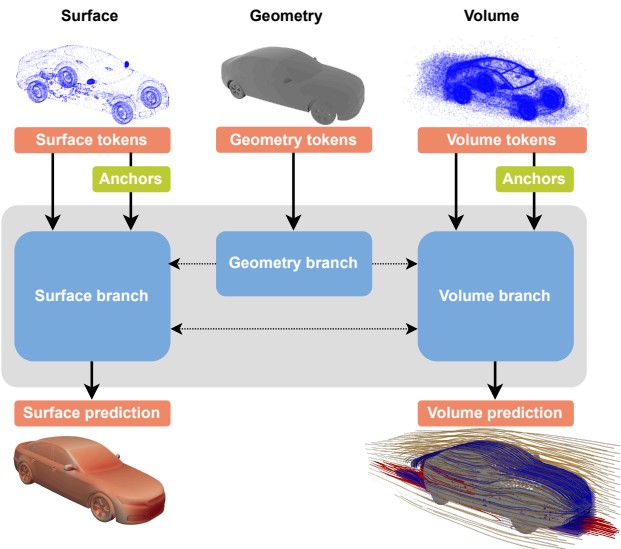

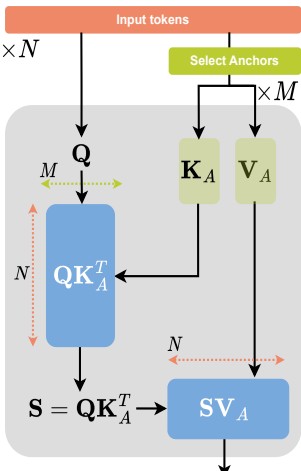

(a) **Multi-branch architecture:** AB-UPT encodes geometries into a reduced set of latent geometry tokens that are integrated into predictions via cross-attention. Interactions between surface and volume fields are modeled via cross-attention.

(b) **Anchor attention:** Physics is approximated using latent anchor tokens that share context via self-attention. Predictions at additional query points cross-attend to anchor tokens.

Figure 3: Overview Anchored-Branched Universal Physics Transformers (AB-UPT).

In recent years, deep neural network-based surrogates have emerged as a computationally efficient alternative in science and engineering (Brunton et al., 2020; Thuerey et al., 2021; Zhang et al., 2023), impacting, e.g., weather forecasting (Pathak et al., 2022; Bi et al., 2023; Lam et al., 2023; Nguyen et al., 2023; Bodnar et al., 2025), protein folding (Jumper et al., 2021; Abramson et al., 2024), or material design (Merchant et al., 2023; Zeni et al., 2025; Yang et al., 2024). In many of these scientific disciplines, the Transformer architecture (Vaswani et al., 2017) has become a central backbone of the recent breakthroughs. Inherently linked to Transformers is the notion of *scale*, both in terms of trainable parameters of the model and the amount of data needed to train those scaled models. However, the notion of scale is ill-defined in the field of neural surrogate modeling for science and engineering, mainly due to the lack of availability of high-volume training data. Hence, we start this work by posing the following question:

> How to define and harness *scale* in the context of (automotive) neural surrogates for CFD simulations?

Scaling large machine learning models is usually framed in terms of the number of trainable parameters and the overall dataset size. Particular to the CFD field, however, is the fact that industry-standard volumetric meshes often exceed more than 100 million cells (i.e., high-dimensional data), while at the same time the amount of simulation samples is low (i.e., low number of independent training samples). Moreover, intricate geometry-dependent interactions between the surface and volumetric fields govern the underlying physics. Further complexity arises from field quantities such as vorticity, which are highly nonlinear yet inherently divergence-free. The combination of high-dimensional non-linear physics phenomena and the low number of industry-standard training samples makes scaling neural surrogate models for CFD particularly hard. Thus, our definition of scale also encompasses *problem scale*, i.e. the fidelity of the simulation (visually represented in Figure 2). To harness problem scale, we propose Anchored-Branched Universal Physics Transformers (AB-UPT) (see Figure 3, a novel modeling scheme, build on the Universal Physics Transformer (UPT) framework, for building neural surrogates for automotive CFD simulations. The modeling is organized into three branches (geometry-encoding, surface-prediction, volume-prediction), with

| Model | Computation unit | Paradigm | Accuracy | Context tokens | Speed (train) | Speed (test) | Simulation from CAD mesh | Neural field decoder |
|---|---|---|---|---|---|---|---|---|
| HRLES | Equations | Numeric | ✓✓✓ | - | - | ✗✗ | ✗ | - |
| RANS | Equations | Numeric | ✓✓* | - | - | ✗ | ✗ | - |
| PointNet | MLPs | Point-based | ✗✗ | ✗ | ✗ | ✓ | ✗ | ✗ |
| GINO | GNO/FNO | Grid interpol. | ✗ | ✗ | ✗ | ✓ | ✓ | ✓ |
| OFormer | Lin. Transformer | Point-based | ✓ | ✗ | ✗ | ✓ | ✓ | ✓ |
| Transolver | Lin. Transformer | Point-based | ✓ | ✗ | ✗ | ✓ | ✗ | ✗ |
| AB-UPT | Transformer | Latent space | ✓✓ | ✓ | ✓ | ✓ | ✓ | ✓ |

Table 1: Conceptual overview of different CFD solvers. Numeric integration schemes, while accurate, are slow and require specialized meshes to converge, with several orders of magnitude variations in fidelity and runtime. Neural surrogates struggle to be trained on the full-resolution mesh, which makes them slow to train or requires multi-GPU setups for million-scale meshes. Next to that, those methods require computation units that favor speed over accuracy. We introduce AB-UPT, a new method incorporating *anchor tokens*. AB-UPT unlocks linear problem scaling, which enables training of neural surrogates on a single GPU in less than a day, even on million-scale meshes. This efficiency also allows us to leverage expressive neural computation units such as the Transformer. Furthermore, our multi-branch design enables simulation with any input mesh, uses reduced latent modeling of physical dynamics. Finally, *anchor attention* acts as a neural field, enabling an efficient divergence-free vorticity field formulation. *It's difficult to make comparisons between RANS simulations and models that are trained on HRLES simulations, since RANS simulations cannot resolve certain phenomena that HRLES can resolve. One example is strong separation.

cross-attention between and self-attention within branches, decoupling feature extraction from field prediction. Within each branch, the encoded simulation points are partitioned into two sets: a small set of selected *anchor* tokens share information among each other via self-attention and together provide context (e.g., about the physical dynamics); the remaining *query* tokens only receive information from the anchor tokens via cross-attention, leading to independent query predictions. This anchor-based design has three critical advantages, resulting in strong generalization performance, inference efficiency, and physics consistency: (i) selecting subsets of points massively increases the amount of effective training instances, and the number of anchor tokens governs the model's computational capacity; (ii) computational cost scales quadratically only with the small number of anchor tokens, while predictions on all other points have linear computational complexity; (iii) interpreting the anchor mechanism as a conditional-field model enables the application of differential operators, and we leverage this capability to derive a divergence-free vorticity-field formulation.

We demonstrate the effectiveness of AB-UPT by comparing its predictive performance against strong surrogate baselines on automotive CFD simulation tasks, including industrial-scale datasets. Key highlights include:

(I) Training models in less than a day, predicting surface and volume fields in seconds on a single GPU.

(II) State-of-the-art predictive accuracy on automotive CFD simulations with up to 140 million mesh cells.

(III) Near-perfect accuracy in drag and lift coefficients computed from pressure predictions.

(IV) Enabling continuous field predictions from CAD geometries, eliminating costly mesh creation.

(V) Accurate and physically consistent vorticity predictions via divergence-free formulation.

## 2 Preliminaries and related work

We outline a conceptual overview and high-level description of our architecture AB-UPT in Table 1. This section provides background information for numerical solvers in Section 2.1 as well as for related work in neural surrogate modeling in Section 2.2. A detailed description of our method follows in Section 3.

### 2.1 CFD for automotive aerodynamics

**Computational fluid dynamics**. Automotive aerodynamics is centered around computational fluid dynamics (CFD) (Versteeg & Malalasekera, 2007; Hirsch, 2007; Pletcher et al., 2012), which is deeply connected to solving the Navier-Stokes (NS) equations. For automotive aerodynamics simulations, the assumptions of incompressible fluids due to low local Mach numbers, i.e., low ratio of flow velocity to the speed of sound, are justified (Ashton et al., 2024b). Thus, the simplified incompressible form of the NS equations (Temam, 2001) is applicable, which conserves momentum and mass of the flow field $\boldsymbol{u}(t, x, y, z) : [0, T] \times \mathbb{R}^3 \to \mathbb{R}^3$ via:

$$\frac{\partial \boldsymbol{u}}{\partial t} = -\boldsymbol{u} \cdot \nabla \boldsymbol{u} + \mu \nabla^2 \boldsymbol{u} - \nabla p + \boldsymbol{f} \ , \quad \nabla \cdot \boldsymbol{u} = 0 \ . \tag{1}$$

To compute a numerical solution, it is essential to discretize the computational domain. In CFD, the finite volume method (FVM) is one of the most widely used discretization techniques. FVM partitions the computational domain into discrete control volumes using a structured or unstructured mesh. The initial geometric representation, typically provided as a computer-aided design (CAD) model in formats such as STL, must be transformed into a volumetric simulation mesh. This meshing process precisely defines the simulation domain, allowing the representation of complex flow conditions, such as those in wind tunnel configurations or open-street environments.[1] Note that meshing highly depends on the turbulence modeling and the flow conditions (Versteeg & Malalasekera, 2007).

**Turbulence modeling in CFD**. Turbulence arises when the convective forces $\boldsymbol{u} \cdot \nabla \boldsymbol{u}$ dominate over viscous forces $\mu \nabla^2 \boldsymbol{u}$, typically quantified by the Reynolds number. Turbulent flows are characterized by a wide range of vortices across scales, with energy cascading from larger structures to smaller ones until viscous dissipation converts it into thermal energy at the Kolmogorov length scale. Although direct numeric simulation (DNS) can theoretically resolve the turbulent flow field by directly solving the NS equations, it requires capturing all scales of motion down to the Kolmogorov scale. This implies extremely high requirements on the discretization mesh, which results in infeasible computational costs for full industrial cases (Versteeg & Malalasekera, 2007).

Therefore, engineering applications rely on turbulence modeling approaches that balance accuracy and computational efficiency. Reynolds-Averaged Navier-Stokes (RANS) (Reynolds, 1895; Alfonsi, 2009) and Large-Eddy Simulations (LES) (Lesieur et al., 2005) are two methods for modeling turbulent flows, each with distinct characteristics. RANS decomposes flow variables into mean and fluctuating components and solves the time-averaged equations, using turbulence models like k-$\epsilon$ (Launder & Spalding, 1974) to account for unresolved fluctuations. While computationally efficient, RANS may lack accuracy in capturing complex or unsteady flows, particularly in cases involving flow separation, where turbulence models are often less effective. In contrast, LES resolves eddies down to a cut-off length and models sub-grid scale effects and their impact on the larger scales. LES offers higher accuracy in capturing unsteady behavior and separation phenomena at the cost of more compute. In cases where LES is too costly, hybrid models like, Hybrid RANS-LES (HRLES) models (Spalart et al., 2006; Chaouat, 2017; Heinz, 2020; Ashton et al., 2022) are an alternative. These models reduce computational demand by using LES away from boundary layers, and RANS near surfaces, where sufficiently resolving the flow in LES would require very high resolution. The DrivAerML (Ashton et al., 2024b) is currently the dataset with the highest fidelity that is publicly available. The dataset utilizes an HRLES turbulence model and runs CFD simulations on 140 million volumetric cells. On the other hand, the DrivAerNet dataset (Elrefaie et al., 2024a;b) runs CFD simulations on up to 23 million volumetric mesh cells with low-fidelity RANS methods, but the dataset has more diverse geometry variations.

**Quantities of interest**. Interesting quantities for automotive aerodynamics comprise quantities on the surface of the car, in the volume around the car, as well as integral quantities such as drag and lift coefficients. The force acting on an object in an airflow is given by

$$\boldsymbol{F} = \oint_S -(p - p_\infty)\boldsymbol{n} + \boldsymbol{\tau}_w dS \ , \tag{2}$$

with the aerodynamic contribution, consisting of surface pressure $p$ and pressure far away from the surface $p_\infty$ times surface normals $\boldsymbol{n}$, and the surface friction contribution $\boldsymbol{\tau}_w$. For comparability between designs,

---

[1]We emphasize the differentiation between raw *geometry mesh* and the re-meshed *simulation mesh* for CFD modeling.

dimensionless numbers as drag and lift coefficients

$$C_\text{d} = \frac{2\,\boldsymbol{F} \cdot \boldsymbol{e}_\text{flow}}{\rho\,v^2 A_\text{ref}},\ C_\text{l} = \frac{2\,\boldsymbol{F} \cdot \boldsymbol{e}_\text{lift}}{\rho\,v^2 A_\text{ref}} \tag{3}$$

are used (Ashton et al., 2024b), where $\boldsymbol{e}_\text{flow}$ is a unit vector into the free stream direction, $\boldsymbol{e}_\text{lift}$ a unit vector into the lift direction perpendicular to the free stream direction, $\rho$ the density, $v$ the magnitude of the free stream velocity, and $A_\text{ref}$ a characteristic reference area. Predicting these surface integrals allows for an efficient estimation when using deep learning surrogates, since these surrogates can directly predict the surface values without the need to model the full 3D volume field, as required by numerical CFD simulations.

**Conserved quantities**. The vorticity field, defined as the curl of the velocity field, i.e., $\boldsymbol{\omega} = \nabla \times \boldsymbol{u}$, is inherently divergence-free due to the mathematical identity that the divergence of a curl is always zero, i.e., $\nabla \cdot \boldsymbol{\omega} = \nabla \cdot \nabla \times \boldsymbol{u} = 0$. However, it is crucial to distinguish the divergence-free property of the vorticity field from the incompressible formulation of the Navier–Stokes equations, where the divergence-free condition applies directly to the velocity field as a consequence of mass conservation in an incompressible (constant-density) fluid, i.e., $\nabla \cdot \boldsymbol{u} = 0$. In the context of the vorticity formulation, the divergence-free condition arises from the definition of vorticity and does not imply or enforce incompressibility of the underlying velocity field.

## 2.2 Transformer blocks for building neural surrogates

**Neural surrogates**. Many neural surrogate models are formulated in the neural operator learning paradigm (Li et al., 2020; 2021; Lu et al., 2021; Seidman et al., 2022; Alkin et al., 2024a;b). In this framework, neural networks represent operators that map between Banach spaces $\mathcal{I}$ and $\mathcal{O}$ of functions defined on compact domains $\mathcal{X}$ and $\mathcal{Y}$, i.e., $\mathcal{G} : \mathcal{I} \to \mathcal{O}$. Neural operators provide continuous outputs that remain consistent across varying input sampling resolutions and patterns. Training a neural operator involves constructing a dataset of input-output function pairs evaluated at discrete spatial locations.

AB-UPT is a neural operator that can encode different input representations (e.g., simulation surface mesh, geometry mesh) via the flexible multi-branch architecture and produce outputs of arbitrary resolution via *anchor attention.*

**Self-attention and cross-attention**. Scaled dot-product attention (Vaswani et al., 2017) is defined upon query $\mathbf{Q}(\mathbf{Z}) \in \mathbb{R}^{N \times d}$, key $\mathbf{K}(\mathbf{Z}) \in \mathbb{R}^{N \times d}$ and value $\mathbf{V}(\mathbf{Z}) \in \mathbb{R}^{N \times d}$ matrices which are linear projections from a latent representation $\mathbf{Z} \in \mathbb{R}^{N \times d}$, written in matrix representation as $\text{Attention}(\mathbf{Q}, \mathbf{K}, \mathbf{V}, \mathbf{Z}) = \text{softmax}(\mathbf{Q}(\mathbf{Z})\mathbf{K}(\mathbf{Z})^T/\sqrt{d})\mathbf{V}(\mathbf{Z})$, where $N$ is the number of tokens and $d$ is the hidden dimension.

Cross-attention is a variant of scaled dot-product attention where a second latent representation $\mathbf{Z}_{KV} \in \mathbb{R}^{M \times d}$ is used as input to $\mathbf{K}(\cdot)$ and $\mathbf{V}(\cdot)$. Intuitively, this formulation incorporates information from $\mathbf{Z}_{KV}$ into $\mathbf{Z}$. It is commonly used to enable interactions between different modeling components (e.g., (Vaswani et al., 2017)), data types (e.g., (Alayrac et al., 2022)), or to compress/expand representations (e.g., (Jaegle et al., 2021)). Roughly speaking, the first two use-cases use $N \approx M$ whereas the third use-case uses $N \ll M$ (compression) or $M \ll N$ (expansion).

AB-UPT uses self-attention throughout the whole model to exchange global information between tokens, cross-attention with geometry tokens as keys/values to incorporate geometry information into the surface/volume branches, and cross-attention between branches to enable surface-volume interactions.

**Neural field output decoding via cross-attention**. Neural fields (Mildenhall et al., 2020) are neural networks that take continuous coordinates as inputs and produce a prediction for this exact coordinate. The continuous nature of the input coordinates enables evaluating this function at arbitrary positions, making it a field. By using embedded coordinates as $\mathbf{Z}$ and a latent representation of, e.g., the simulation state as $\mathbf{Z}_{KV}$, cross-attention acts as a conditional neural field (Wang et al., 2024b) as each query vector retrieves information from the latent representation to produce a prediction for a given query coordinate. It is worth noting that the neural field decoding via cross-attention operates on a point-wise basis and hence, is independent of the number of query points.

Neural field decoding approaches have been commonly utilized in the context of neural surrogates (Li et al., 2023a; Alkin et al., 2024a; Wang et al., 2024b) to produce high-resolution outputs from a latent representation.

AB-UPT also utilizes a neural field decoder via cross-attention. Contrary to previous methods, our decoder can use a more expressive context as expressive self-attention is employed between anchor tokens, where all anchor tokens (including intermediate representations) serve as a condition for the neural field decoder.

## 2.3 Related work

In this section, we will discuss work related to neural surrogate modeling and hard physical constraints in neural networks.

**Reduced latent space modeling**. Training neural surrogates directly on a complete simulation mesh can be computationally expensive and, in some cases, even infeasible. To address this issue, compression of the input mesh into a reduced latent space can be employed. Next, a decoder (often functioning as a point-wise decoder) needs to be able to reconstruct the output mesh conditioned on the compressed latent representation and single query points. To satisfy this requirement, the decoder must function as a conditional neural field. Such approaches are introduced in AROMA (Serrano et al., 2024), CViT (Wang et al., 2024b), Knigge et al. (2024), GOAT (Wen et al., 2025), LNO (Wang & Wang, 2024), and UPT (Alkin et al., 2024a). Moreover, UPT additionally introduces a patch-embedding analogue for compressing general geometries via supernode pooling.

**Unreduced physics modeling**. Methods like Reg-DGCNN (Wang et al., 2019; Elrefaie et al., 2024a;b), Graph-UNet (Gao & Ji, 2019), PointNet (Qi et al., 2017a), and Transformer models with linear complexity, such as Erwin (Zhdanov et al., 2025), FactFormer (Li et al., 2024a), GNOT (Hao et al., 2023), ONO (Xiao et al., 2023), OFormer (Li et al., 2023a) or Transolver (Wu et al., 2024) have constant compute complexity w.r.t. the number of input points. This allows such models to process the unreduced simulation mesh at high computation costs, as shown in Transolver++ (Luo et al., 2025), which demonstrates processing meshes with 2.5 million cells via sequence parallelism on four GPUs. Although technically feasible, such approaches also become infeasible at larger scales. For example, linearly extrapolating the reported GPU requirements of Transolver++ would necessitate roughly 256 GPUs to process a simulation mesh with 140 million cells. In contrast, AB-UPT can process simulations of 140 million cells on a single GPU.

**Decoupling of encoder and decoder**. Decoupling encoding and decoder components is often beneficial for tasks like predicting 3D flow fields directly from geometric inputs (see e.g., GINO (Li et al., 2023b), GOAT (Wen et al., 2025), LNO (Wang et al., 2024a), OFormer (Li et al., 2023a), and UPT). Additionally, a decoupled latent space representation allows for scalable decoding to a large number of output mesh cells since such models can cache encoded latent states and decode output queries in parallel. Moreover, decoupling the encoder and decoder also allows for independent input and output meshes. This means the model does not require the same simulation mesh used for ground truth output as its encoder input during training (i.e., the encoder can be simulation mesh independent), providing greater flexibility in data handling.

**Hard physical constraints in neural networks**. A popular approach to embed physical principles into neural networks is to use the governing equations as a *soft constraint*, penalizing the partial differential equation (PDE) residuals in the loss function (Raissi et al., 2019; Li et al., 2024b). This encourages but does not strictly enforce physical consistency, and often requires second-order methods to alleviate optimization challenges (Wang et al., 2025). An alternative is to enforce *hard constraints*, for instance by projecting the output of an unconstrained model onto a physically feasible solution space (Négiar et al., 2023; Hansen et al., 2023; Chalapathi et al., 2024; Utkarsh et al., 2025). In contrast, our work enforces physical principles such as the divergence-free nature of the vorticity field as a *hard constraint* by construction. Our approach is closely related to the work by Richter-Powell et al. (2022) and Liu et al. (2024), who construct divergence-free networks using differential operators. However, to our knowledge, none of the aforementioned physics-informed approaches have been scaled to the massive model, data, and problem sizes considered in this work.

# 3 Anchored-Branched Universal Physics Transformers

We introduce Anchored-Branched Universal Physics Transformers (AB-UPT), which leverages two key modeling components: (i) the *multi-branch* architecture outlined in Section 3.1 separates geometry encoding, surface-level simulation, and volume-level simulation into distinct branches that interact with each other;

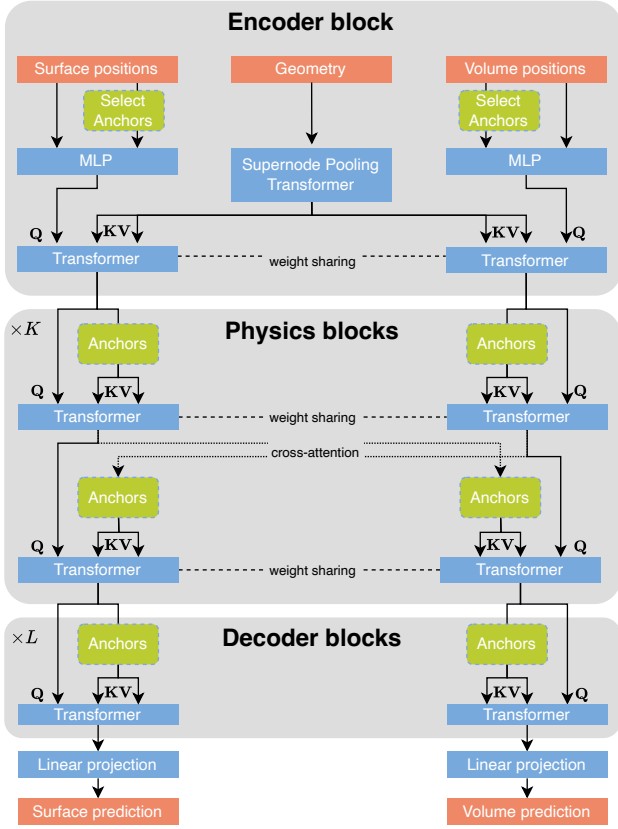

Figure 4: The Anchored-Branched Universal Physics Transformers model is a multi-branch Transformer neural operator using three distinct branches: a surface, geometry, and volume branch. The physics blocks within each branch share parameters and interact through cross-attention. To handle problems at industry-scale, we select a subset of $M$ points from the input point cloud. We call these points *anchor tokens*, and they serve as the keys and values for attention computation. This approach significantly reduces the complexity of attention, making it possible to scale our model to industry-standard simulation meshes. Query positions define where physical quantities are predicted. For simplicity, we assume anchors and queries are disjoint subsets of all positions, enabling evaluation at anchor locations. We will demonstrate in Section 4.6 that anchors can also be placed arbitrarily (e.g., on a regular grid), in which case they may fall in regions where evaluating physical quantities is not physically meaningful (e.g., points inside a car geometry).

(ii) *anchor attention* described in Section 3.2 enables modeling of complex physical dynamics by leveraging expressive dot-product attention (Vaswani et al., 2017) within a heavily reduced set of *anchor tokens*. To generate predictions for the full simulation mesh during inference, *anchor attention* employs cross-attention between (millions) of *query tokens* and the small amount of *anchor tokens*, which has linear inference complexity. Additionally, *anchor attention* can be seen as a conditional neural field, which allows creating arbitrary output resolutions, and enables the application of differential operators to enforce hard physical constraints such as divergence-free vorticity-field predictions (see Section 3.3).

## 3.1 Multi-branch architecture

Our model builds on the Universal Physics Transformer (UPT) (Alkin et al., 2024a) and multi-branch Transformers for neural operators (Alkin et al., 2024b; Esser et al., 2024). Multi-branch Transformers consist

of multiple branches (which can share parameters) that each handle different data streams or modalities. We define a geometry branch, a surface branch, and a volume branch (see Figures 3 and 4 for an overview). Those branches interact differently in respective encoding, physics, and decoding blocks, and allow us to enable scalability to high-resolution outputs via the new *anchor attention*. We distinguish the geometry, surface, and volume branches by the superscripts $g$, $s$, and $v$, respectively.

**Multi-branch encoder block**. The geometry branch encodes an arbitrary input geometry $\mathcal{M}$ (which can be different than the simulation mesh), from a simulation sample, which is represented as a finite discretized point cloud, consisting of $N^g$ points in a 3D space, i.e., $\mathbf{X}^g \in \mathbb{R}^{N^g \times 3}$. The 3D coordinates are embedded using the Transformer positional encoding (Vaswani et al., 2017). We found that no additional input features are required for any of the branches. However, we note that it is possible to add additional inputs such as surface normals, signed distance function (SDF) features, or simulation design parameters. Following the approach of Alkin et al. (2024a), a supernode pooling block $\mathcal{S}$ maps the input geometry $\mathcal{M}$ into a reduced set of supernode representations $\mathbf{S}$, which capture local geometry information within some radius $r$ of the supernode:

$$\mathcal{S} : \mathbf{X}^g \in \mathcal{M} \xrightarrow{\text{Embed}} \mathbf{H}^g \in \mathbb{R}^{N^g \times d} \xrightarrow{\text{Supernode pooling}} \mathbf{S} \in \mathbb{R}^{S \times d}, \tag{4}$$

where $d$ is the dimensionality of the model's latent representations. Distances in the supernode pooling are computed only based on the input coordinates: first, we sample a subset of $S$ *supernodes*, where typically $S \ll N^g$, from the coordinates $\mathbf{X}^g$. Through a Graph Neural Operator (GNO) layer (Li et al., 2020), we aggregate information from neighboring coordinates within a radius $r$. The resulting supernode representations $\mathbf{S}$ are passed through a self-attention block, allowing the supernodes to aggregate global information from the other supernodes, resulting in the output of the geometry branch

$$\mathcal{G} : \mathbf{S} \xrightarrow{\text{Attention}} \mathbf{Z}^g \in \mathbb{R}^{S \times d}. \tag{5}$$

We discretize the input of the surface and volume branches by turning the simulation mesh into sets of 3D point clouds. Specifically, we select $N^s$ points from the surface and $N^v$ points in the 3D volume, i.e., $\mathbf{X}^s \in \mathbb{R}^{N^s \times 3}$ and $\mathbf{X}^v \in \mathbb{R}^{N^v \times 3}$. To embed the 3D coordinates, we use the Transformer positional encoding as in the geometry branch. Then, we encode these embedded points by using two multi-layer perceptron (MLP) encoders, $\mathcal{E}^s$ and $\mathcal{E}^v$. We employ separate MLP encoders for surface coordinates ($\mathcal{E}^s$) compared to volume coordinates ($\mathcal{E}^v$) to provide explicit contextual information, designating which points reside on the surface and within the volume. To summarize, the surface and volume encoders are defined as follows:

$$\mathcal{E}^s : \mathbf{X}^s \xrightarrow{\text{Embed}} \mathbf{H}^s \xrightarrow{\text{MLP}} \mathbf{Z}^s \in \mathbb{R}^{N^s \times d}, \tag{6a}$$

$$\mathcal{E}^v : \mathbf{X}^v \xrightarrow{\text{Embed}} \mathbf{H}^v \xrightarrow{\text{MLP}} \mathbf{Z}^v \in \mathbb{R}^{N^v \times d}. \tag{6b}$$

**Multi-branch physics blocks and decoder**. To bridge the geometry and the surface/volume branch, we use a cross-attention block, $\mathcal{P}^s$ and $\mathcal{P}^v$, which shares parameters among the surface and volume branches.

$$\mathcal{P}^s : \left( \mathbf{Q}(\mathbf{Z}^s), \mathbf{K}(\mathbf{Z}^g), \mathbf{V}(\mathbf{Z}^g) \right) \xrightarrow{\text{Attention}} \mathbf{Z}_0^s, \tag{7a}$$

$$\mathcal{P}^v : \left( \mathbf{Q}(\mathbf{Z}^v), \mathbf{K}(\mathbf{Z}^g), \mathbf{V}(\mathbf{Z}^g) \right) \xrightarrow{\text{Attention}} \mathbf{Z}_0^v, \tag{7b}$$

where we write $\mathbf{Q}(\mathbf{Z})$, $\mathbf{K}(\mathbf{Z})$ and $\mathbf{V}(\mathbf{Z})$ to denote the linear transformation from representation $\mathbf{Z}$ to its corresponding query, key and value matrix. The queries are the encoded surface and volume points, and the output of the geometry branch serves as keys and values. This setup enables the surface and volume points to effectively incorporate information from the geometry encoder, providing global information about the input geometry.

The surface and volume branch consists of $K$ physics blocks. Each physics block $k$, where $k \in \{1, \ldots, K\}$, consists of a self-attention and cross-attention block, which share parameters between the surface and volume branches. Each physics block starts with a self-attention block, where the queries, keys, and values are all

derived from the output $\mathbf{Z}_{k-1}$ of the preceding physics block $k-1$, where $\mathbf{Z}_0$ from Equation (7) is the input to the first block. Next is a cross-attention block, enabling surface-volume interaction. For the surface branch, the queries originate from the previous block within the surface branch. However, the keys and values come from the volume branch $k$. For the volume branch, however, it is reversed: the queries come from the volume branch, and the keys and values from the surface branch. These cross-attention layers allow for intricate interactions between the surface and volumetric fields. Formally, physics block $k$ for the surface and volume branch, $\mathcal{B}_k^s$ and $\mathcal{B}_k^v$, are defined by:

$$\mathcal{B}_k^s : \left\{ \mathbf{Q}(\mathbf{Z}_{k-1}^s), \mathbf{K}(\mathbf{Z}_{k-1}^s), \mathbf{V}(\mathbf{Z}_{k-1}^s) \right\} \xrightarrow{\text{Attention}} \mathbf{Z}_k^{\prime s} \to \left\{ \mathbf{Q}(\mathbf{Z}_k^{\prime s}), \mathbf{K}(\mathbf{Z}_k^{\prime v}), \mathbf{V}(\mathbf{Z}_k^{\prime v}) \right\} \xrightarrow{\text{Attention}} \mathbf{Z}_{k+1}^s, \quad (8a)$$

$$\mathcal{B}_k^v : \left\{ \mathbf{Q}(\mathbf{Z}_{k-1}^v), \mathbf{K}(\mathbf{Z}_{k-1}^v), \mathbf{V}(\mathbf{Z}_{k-1}^v) \right\} \xrightarrow{\text{Attention}} \mathbf{Z}_k^{\prime v} \to \left\{ \mathbf{Q}(\mathbf{Z}_k^{\prime v}), \mathbf{K}(\mathbf{Z}_k^{\prime s}), \mathbf{V}(\mathbf{Z}_k^{\prime s}) \right\} \xrightarrow{\text{Attention}} \mathbf{Z}_{k+1}^v. \quad (8b)$$

The final decoder block consists of $L$ self-attention blocks that do not share parameters and hence operate independently. The query, key, and value representations all originate from the previous layer, and, therefore, there is no interaction anymore between the two branches. This design allows for branch-specific modeling, meaning each branch can refine the output predictions independently, without interaction with the other branches. Finally, a single linear projection is applied as a decoder to generate the final output predictions for each branch.

We use a standard pre-norm Transformer architecture akin to Vision Transformers (Dosovitskiy et al., 2021) where each attention block is followed by an MLP block (which we omit above for notational simplicity). Additionally, we employ Rotary Position Embeddings (RoPE) (Su et al., 2024) in all attention blocks.

## 3.2 Anchor attention

High-fidelity numerical simulations in industry often rely on extremely fine mesh discretizations, sometimes comprising hundreds of millions of cells (Ashton et al., 2024b). In practice, representing such large point clouds (meshes) as inputs to surrogate models becomes computationally challenging. In particular, Transformer models (Vaswani et al., 2017)—the workhorse of many modern deep learning architectures—capture dependencies among all $N$ input tokens via self-attention:

$$\text{Attention}(\mathbf{Q}(\mathbf{Z}), \mathbf{K}(\mathbf{Z}), \mathbf{V}(\mathbf{Z})) = \text{softmax}\left( \frac{\mathbf{Q}(\mathbf{Z})\mathbf{K}(\mathbf{Z})^\top}{\sqrt{d}} \right) \mathbf{V}(\mathbf{Z}) .$$

Because forming the full $\mathbf{Q}(\mathbf{Z})\mathbf{K}(\mathbf{Z})^\top$ matrix takes $\mathcal{O}(N^2)$ time and memory, this operation becomes infeasible when $N$ is on the order of hundreds of millions.

To mitigate this problem, we introduce the concept of *anchor attention*, which splits the discretized input mesh (i.e., the point cloud) into two parts: (i) a small subset $\mathbf{A}$ of $M$ *anchor tokens* is uniformly sampled from the $N$ input tokens and used to derive the key matrix $\mathbf{K}(\mathbf{A})$ and value matrix $\mathbf{V}(\mathbf{A})$ for the attention mechanism; (ii) the query matrix $\mathbf{Q}(\mathbf{Z})$, which is derived from *all* $N$ input tokens, then cross-attend to the anchor key and value matrices. Intuitively, this formulation splits the $N$ input tokens into $M$ *anchor tokens* (used as queries, keys, and values in the attention) and $N - M$ *query tokens* (used only as queries in the attention). It is equivalent to self-attention within the $M$ *anchor tokens* and cross-attention between $N - M$ *query tokens* and the $M$ *anchor tokens*. Formally, anchor attention is defined as follows:

$$\mathcal{I}_A \sim \text{UniformSample}(\{1, 2, \ldots, N\}, M) \qquad M \ll N,$$

$$\mathbf{A} = \mathbf{Z}[\mathcal{I}_A] \qquad \in \mathbb{R}^{M \times d},$$

$$\text{AnchorAttention}(\mathbf{Q}(\mathbf{Z}), \mathbf{K}(\mathbf{A}), \mathbf{V}(\mathbf{A})) = \text{softmax}\left( \frac{\mathbf{Q}(\mathbf{Z})\mathbf{K}(\mathbf{A})^\top}{\sqrt{d}} \right) \mathbf{V}(\mathbf{A}) \qquad \in \mathbb{R}^{N \times d}, \quad (9)$$

where $\mathcal{I}_A$ is sampled once for every sample in a forward pass.

For notational simplicity, we assume that anchor and query tokens come from the same set of points (e.g., positions from a CFD mesh). However, anchor and query tokens can also be obtained from completely

different sets. We will demonstrate this in Section 4.6 where anchor tokens are sampled from a CAD geometry (surface anchors) or a regular grid (volume anchors) and the query tokens are obtained from the CFD mesh (both surface and volume query tokens).

This design allows us to *focus on problem scale* (c.f. Figure 2): by choosing the number of anchor tokens $M$, we directly control the model's computational capacity, while enabling pointwise predictions at arbitrary points in the continuous output domain. These anchor tokens span both the surface and volume mesh, and their selection is crucial for capturing problem-specific structure. Increasing $M$ enhances contextual information available at each query location without changing the amount of model parameters, thereby improving performance up to a saturation point (see Figure 5).

Importantly, the $N - M$ query tokens only need to compute attention with a much smaller set of $M$ key-value pairs (derived from the anchor tokens) compared to full self-attention, which computes all $N^2$ pairs. Anchor attention therefore *reduces the computational complexity* of the attention layers from $\mathcal{O}\left(N^2\right)$ to $\mathcal{O}\left(M^2 + (N - M)M\right) = \mathcal{O}\left(MN\right)$, where $M^2$ is the self-attention between the $M$ anchor tokens and $(N - M)M$ is the cross-attention from $N - M$ query tokens to $M$ anchor tokens. This is crucial because surrogate models often need to be evaluated at a large number of input points (e.g., to obtain accurate drag/lift coefficient predictions as shown in Figure 1).

Additionally, since there are no interactions between query tokens, an arbitrary number of query tokens can be used per forward pass. Therefore, query tokens can be processed in chunks, enabling constant memory consumption by using a constant number of queries. When processing all $N$ input tokens at once (*end-to-end inference*), the memory complexity of anchor attention is $\mathcal{O}(N)$ (we use an efficient attention implementation (Dao et al., 2022)). However, as $N$ can be $> 100$ million, it would still run into out-of-memory errors on a single GPU. By chunking the $N - M$ query tokens into chunks of size $C$, memory complexity reduces to $\mathcal{O}(M + C)$, i.e., constant memory consumption. We refer to this methodology as *chunked inference* (c.f. inference memory in Figure 1). Moreover, in chunked inference, the key-value pairs of anchor tokens can be cached when calculating them in the first chunked forward pass to omit recalculating them for subsequent chunks (anchor tokens remain constant for all chunks). However, as $M \ll N$, this is often negligible.

Finally, by treating the anchors as conditioning set for the queries, anchor attention turns the architecture into a *conditional field model*: the field values at any query location are predicted from the query location itself and the context provided by the anchors, but independently from any other query location. This has two important benefits. First, this enables creating predictions at arbitrary query positions, which allows AB-UPT to predict at arbitrary resolution. Many simulation insights can be obtained from much lower resolutions than necessary for numeric simulation (e.g., drag/lift coefficients in Figure 6). Second, it allows us to impose physics-based constraints directly into the model predictions, such as a divergence-free condition for vorticity (see Section 3.3). In the context of Neural Processes, the concept of conditioning the output prediction on a (small) subset of context has been explored in (Feng et al., 2023; Ashman et al., 2024).

### 3.3 Enforcing physics consistency

**Anchored neural field enables derivative operators**. As discussed in Section 2.1, physical fields are often intrinsically governed by local differential operators, enforcing constraints that reflect fundamental conservation laws. However, imposing differential-operator constraints on neural networks requires a field-based formulation, allowing *pointwise* derivative computation. Conventional Transformer architectures cannot be directly viewed as continuous fields because they map input sets to output sets. By contrast, as described in Section 3.2, our anchored neural field architecture treats the network as a conditional field, using anchor tokens' key/value vectors as the context to produce pointwise predictions at arbitrary query positions. This approach enables the application of local differential operators. To demonstrate and leverage the potential of the anchored-field formulation, we model the vorticity as an intrinsically divergence-free field, constructed from the local rotation of the velocity (like the real physical field), thus ensuring physical consistency.

**Divergence-free vorticity field**. Let $\boldsymbol{u} : \mathbb{R}^3 \to \mathbb{R}^3$ be the velocity field and denote its value at any point $\boldsymbol{x} = (x_1, x_2, x_3)$ by $\boldsymbol{u}(\boldsymbol{x})$. The local rotation of the velocity field is characterized by the *vorticity*, which is

defined as the curl of the velocity:

$$\boldsymbol{\omega}(\boldsymbol{x}) = \nabla \times \boldsymbol{u}(\boldsymbol{x}) = \begin{pmatrix} \frac{\partial u_3}{\partial x_2} - \frac{\partial u_2}{\partial x_3} \\ \frac{\partial u_1}{\partial x_3} - \frac{\partial u_3}{\partial x_1} \\ \frac{\partial u_2}{\partial x_1} - \frac{\partial u_1}{\partial x_2} \end{pmatrix}. \tag{10}$$

A well-known mathematical property of the vorticity (and more generally the curl of a field) is that its divergence vanishes, i. e., $\nabla \cdot (\nabla \times \boldsymbol{u}) = 0$. Intuitively, this means that vorticity has no monopole-like sources or sinks: vortex lines can only form closed loops or extend to infinity, never beginning or ending at a point.

Models that directly predict the vorticity field $\hat{\boldsymbol{\omega}}(\boldsymbol{x})$ without further structure cannot guarantee $\nabla \cdot \hat{\boldsymbol{\omega}} = 0$, and thus lead to "unphysical" predictions that introduce spurious sources and sinks of vorticity. Physics-Informed Neural Networks (PINN) typically address this by incorporating the residuals of the PDE and boundary conditions into the loss function as *soft constraints*. By contrast, in this work we show that we can enforce the divergence-free condition as an architectural *hard constraint* by explicitly parameterizing the vorticity field as $\hat{\boldsymbol{\omega}}(\boldsymbol{x}) = \nabla \times \hat{\boldsymbol{u}}(\boldsymbol{x})$, where $\hat{\boldsymbol{u}}$ is the modeled velocity field. In other words, we define the vorticity as the curl of the modeled velocity field, which guarantees $\nabla \cdot \hat{\boldsymbol{\omega}} = 0$ by construction.

**Finite-difference curl approximation**. We found that computing the spatial derivatives in Equation (10) via PyTorch's automatic differentiation is inefficient due to suboptimal support of vectorized cross-attention in `torch.compile`. We therefore instead compute the spatial derivatives with a central-difference approximation,

$$\frac{\partial \hat{u}_i}{\partial x_j}(\boldsymbol{x}, \cdot) \approx \frac{\hat{u}_i(\boldsymbol{x} + \delta\,\boldsymbol{e}_j, \cdot) - \hat{u}_i(\boldsymbol{x} - \delta\,\boldsymbol{e}_j, \cdot)}{2\,\delta}, \tag{11}$$

where $\hat{u}_i$ is the i-th (scalar) velocity output, $\boldsymbol{e}_j$ is the unit vector in the $x_j$-direction, and $\delta$ is the step-size. Here, the dot in $\hat{\boldsymbol{u}}(\boldsymbol{x}, \cdot)$ indicates that the prediction also depends on additional inputs such as the anchor token's key and value vectors, which we omit for notational simplicity. To approximate the partial derivatives w. r. t. each of the three spatial coordinates, we perform six additional forward passes, each applying an offset $+\delta\boldsymbol{e}_j$ and $-\delta\boldsymbol{e}_j$ to the input position.

**Correcting for data normalization**. In order to effectively train neural networks, input and target data is usually shifted and scaled into some appropriate range. Since the vorticity is defined through spatial derivatives of velocities w. r. t. the input coordinates in the raw *physics space*, the effect of scaling must be taken into account. This is because we compute the spatial derivatives of the neural network, i. e. in *network space*. Hence, when scaling the 3D input positions by a vector $\boldsymbol{a}$ (from physics to network space) and the outputs by vector $\boldsymbol{b}$ (from network to physics space), then the $3 \times 3$ network-space Jacobian matrix of partial derivatives must be multiplied by the outer-product matrix $\boldsymbol{b}\boldsymbol{a}^T$, resulting in the physics-space Jacobian.

## 4 Experiments

We begin our experiments by showing that neural surrogates do not need to process the full mesh to obtain accurate models (Section 4.2), which enables efficient training on large problem-scales. Using this insight, we design Anchored-Branched Universal Physics Transformers (AB-UPT) by adapting the UPT architecture (Alkin et al., 2024a) to handle complicated physical dynamics of CFD simulations with million-scale meshes (Section 4.3). To demonstrate the effectiveness of AB-UPT, we compare it against other neural surrogate models in Section 4.4. Next, we demonstrate how AB-UPT can tackle practical use-cases such as calculating aerodynamic drag and lift coefficients (Section 4.5), simulating directly from a CAD geometry at inference time without the expensive creation of a simulation mesh (Section 4.6) as well as enforcing physical consistency via a divergence-free vorticity formulation (Section 4.7). Finally, we show scaling behavior of AB-UPT w.r.t. problem, model, and data scales in Section 4.8. Additional model design ablations and benchmarks are provided in Appendix A and B. The experimental setup for AB-UPT is outlined in Appendix C.3. The inference code for AB-UPT is available at: https://github.com/Emmi-AI/AB-UPT.

Unless stated otherwise, we train AB-UPT in fp16 mixed precision, circumventing previously reported instabilities (Alkin et al., 2024a) via fp32 positional embeddings and RoPE (Su et al., 2024). See Appendix A.6 for more details.

Table 2: Overview of considered datasets. Datasets vary in the number of surface/volume points (#Points) and their variables. High-quality datasets contain surface pressure $p_s$ and wallshearstress $\tau$ as well as volume pressure $p_v$, velocity $u$ and vorticity $\omega$. As CFD simulations are expensive to compute and store, these datasets do not contain many simulations, which also need to be split into train/validation/test splits.

| Dataset | Surface | | | Volume | | | | Properties | | |
|---|---|---|---|---|---|---|---|---|---|---|
| | #Points | $p_s$ | $\tau$ | #Points | $p_v$ | $u$ | $\omega$ | #Simulations | Size[2] | Cost[3] |
| ShapeNet-Car | 4K | ✓ | ✗ | 29K | ✗ | ✓ | ✗ | 789/0/100 | 1 GB | <\$1K |
| AhmedML | 1M | ✓ | ✓ | 21M | ✓ | ✓ | ✓ | 400/50/50 | 409 GB | \$200K |
| DrivAerML | 9M | ✓ | ✓ | 142M | ✓ | ✓ | ✓ | 400/34/50 | 2677 GB | \$1M |

## 4.1 Datasets

We consider three computational fluid dynamics datasets for automotive aerodynamics: ShapeNet-Car (Umetani & Bickel, 2018), AhmedML (Ashton et al., 2024a), and DrivAerML (Ashton et al., 2024b). Table 2 outlines properties of these datasets.

## 4.2 Efficient training on million-scale CFD meshes

Numerical simulations require fine-grained meshes with sophisticated meshing algorithms to keep the numerical solver stable and obtain accurate results. Contrarily, neural surrogates, i.e., neural approximations of a numerical simulation, learn a conceptually different way of resolving the physical dynamics in a system. Consequently, we question the necessity of the fine-grained million-scale input mesh for a neural simulation by training neural surrogates with various input resolutions by randomly subsampling the CFD mesh. Figure 5 shows that the accuracy of a neural solver saturates at relatively coarse resolutions, where increasing resolution beyond a dataset-dependent threshold only increases computational costs without (test) accuracy gains. We use this insight to efficiently train neural surrogate models on meshes up to 140 million volume points by drastically reducing the input resolution and modeling physical dynamics in this reduced input space. This allows training models in less than a day on a single NVIDIA H100 GPU.

While such approaches have been explored in the past (Alkin et al., 2024a;b), many neural surrogate architectures focus on linear attention mechanisms (Li et al., 2023a; Wu et al., 2024) or computation parallelization (Luo et al., 2025) to be able to train with millions of inputs. However, such approaches would still be prohibitively expensive. For example, extrapolating the reported GPU requirements for training Transolver++ (Luo et al., 2025) to the 150 million input points from DrivAerML (Ashton et al., 2024b) would necessitate roughly 256 GPUs.

We show in Figure 5 that it is feasible to train standard Transformer self-attention to accurately predict simulation results on million-scale meshes by training on lower resolutions, outperforming linear Transformer architectures (such as Transolver). As self-attention requires quadratically more compute w.r.t. the number of inputs, computational demands increase. However, the number of GPU-hours required to train models is still low (models with 16 thousand surface/volume points train in less than 10 GPU-hours). Given even more complicated physical simulations, it would be no issue to scale our AB-UPT to more GPUs via sequence parallelism (Liu et al., 2023), which distributes tokens among multiple GPUs, to allow processing millions of tokens despite the quadratic complexity of self-attention. Due to the low amount of GPU-hours required, even for physical dynamics that arise from meshes with over 100 million cells, AB-UPT could be easily scaled to even more complicated problems. For reference, foundation models in domains such as computer vision (e.g., Oquab et al. (2023)) use 10 to 100 thousand GPU-hours, and compute requirements for language models can be millions of GPU-hours (e.g., Dubey et al. (2024)).

---

[2]Size denotes the total size of the data necessary for training AB-UPT, which only includes positions and the predicted surface/volume variables. The full raw dataset would be even larger (e.g., $\sim$ 30TB for DrivAerML).

[3]Cost estimated with \$3 per Amazon EC2 `hpc6a.x48xlarge` node-hour which was used for both AhmedML and DrivAerML. This is the publicly visible on-demand hourly rate. Actual costs may vary based instance provisioning model.

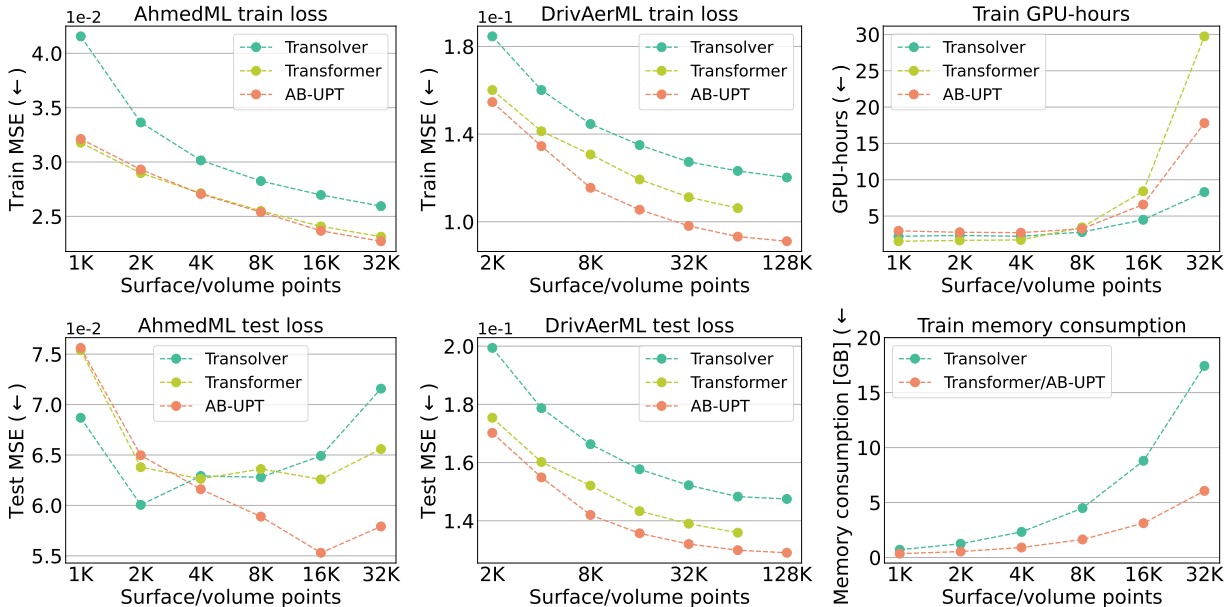

Figure 5: Train and test losses with varying number of training inputs. Contrary to numerical solvers, neural surrogates do not require million-scale meshes to train accurate models. Training on more points beyond a dataset-specific threshold results in overfitting and requires larger datasets (either in terms of data-scale, i.e., number of simulations, or problem-scale, i.e., complexity of the problem) to benefit from the increased modeling capacity when using more points, which is costly. This can be observed in the first and second row, where AhmedML and DrivAerML have the same data scale (400 train simulations), but AhmedML has a smaller problem scale, resulting in overfitting. Note that increasing data- or model-scale is orthogonal to our work. Although runtime increases with a larger number of inputs, the amount of GPU-hours required is still low, and scaling to even larger problem-scales could be easily implemented via sequence parallelism (Liu et al., 2023), which allows scaling quadratic self-attention to millions of tokens. Note that Transformer/AB-UPT requires less memory due to constant memory complexity of FlashAttention (Dao et al., 2022), whereas Transolver uses memory-heavy `torch.einsum` operations to create slices. See Appendix B.5 for more details on Transolver scaling.

Note that random subsampling, as we apply it to create a low-resolution input mesh, is also a form of data augmentation where a network will never see the same constellation of inputs twice during training. As high-quality numerical CFD simulations are expensive to obtain (e. g., DrivAerML costs $1M, see Table 2), datasets often have less than a thousand samples, where this form of regularization helps to prevent overfitting.

## 4.3 Model design

We design AB-UPT, a successor of UPT that preserves useful properties of UPT, such as using a neural field for decoding and being mesh independent (i.e., the input is decoupled from the output). We observe (in Table 3) that UPT is not able to model complicated dynamics such as the vorticity well, and gradually change the UPT architecture to employ techniques that allow efficient scaling to large problem sizes and obtain significantly higher accuracies than a vanilla UPT. We visualize design choices in Table 3 using AhmedML and motivate the design decisions below. Appendix A.1 validates this study on DrivAerML. Additional micro-design choices for the geometry branch are presented in Appendix A.2.

**Baseline**. Starting with a UPT model, we train it on AhmedML (Ashton et al., 2024a) and evaluate surface pressure $p_s$, volume velocity $u$, and volume vorticity $\omega$. UPT struggles with complex decoding tasks (e. g., decoding vorticity) as the decoder consists of a single cross-attention block, and hence, the vast majority of the compute is allocated to encoding.

Table 3: Impact of design steps for transforming UPT into AB-UPT. Performance is measured as relative L2 error (in %) for surface pressure $p_s$, volume velocity $u$ and volume vorticity $\omega$. Train speed measures seconds per 100 update steps with batch size 1. Test speed measures seconds to create a prediction for the full simulation mesh. Speeds are measured on a NVIDIA H100 GPU. The study is conducted on AhmedML.

| | | L2 error (↓) | | | Neural field | Mesh independent | Train speed | Test speed |
|---|---|---|---|---|---|---|---|---|
| | | $p_s$ | $u$ | $\omega$ | | | | |
| baseline | UPT | 4.38 | 3.20 | 37.13 | ✓ | ✓ | 7.0 | 0.2 |
| macro design | + large decoder | 4.25 | 2.73 | 15.03 | ✓ | ✓ | 14.7 | 1.4 |
| | + decoder-only with self-attn | 3.41 | 2.09 | 6.76 | ✗ | ✗ | 14.9 | 163 |
| re-add properties | + anchor attention | 3.41 | 2.09 | 6.76 | ✓ | ✗ | 14.9 | 2.6 |
| | + decouple geometry encoding | 3.31 | 2.09 | 6.64 | ✓ | ✓ | 16.1 | 2.5 |
| micro design | + split surface/volume branch | 3.50 | 2.02 | 6.91 | ✓ | ✓ | 11.0 | 1.5 |
| | + cross-branch interactions | 3.35 | 2.08 | 6.76 | ✓ | ✓ | 11.0 | 1.5 |
| | + branch-specific decoders | 3.01 | 1.90 | 6.52 | ✓ | ✓ | 11.0 | 1.5 |

**Macro design**. We make major changes to the UPT design. First, scaling the decoder to contain equally many parameters as the encoder, which greatly improves performance, at the cost of additional runtime ("+ large decoder"). As scaling the decoder greatly improved performance, we investigate a variant without any encoder by changing the decoder from cross-attending to the encoder tokens to self-attention between all query points ("+ decoder-only with self-attn"). This variant is essentially a plain Transformer without any bells and whistles. While this variant improves performance further, it also loses beneficial properties and is slow in inference due to the quadratic complexity of self-attention.

**Reintroducing properties**. Next, we aim to modify the architecture to reintroduce beneficial properties (neural field decoder, mesh independence, and fast inference) of UPT while preserving the obtained performance gains. To this end, we introduce anchor attention ("+ anchor attention"), which converts the self-attention into a neural field by using self-attention only within a small subset of *anchor tokens*. Other tokens, the *query tokens*, only attend to the anchor tokens. This type of attention is a conditional neural field where positions can be encoded into *query tokens*, which decodes a value for a particular position, without influencing the *anchor tokens*. This allows efficient inference as the quadratic complexity of self-attention is limited to a small number of *anchor tokens*, while creating predictions for query tokens has linear complexity due to employing cross-attention. See Section 3.2 for a detailed formulation of anchor attention.

Further, we reintroduce a light-weight geometry encoding ("+ decouple geometry encoding") by obtaining *geometry tokens* from a light-weight UPT encoder (supernode pooling → 1 Transformer block, see equations 4 and 5). The first self-attention block is additionally replaced by a cross-attention to the *geometry tokens* to extract geometry information for every anchor/query token. Since the geometry encoder is not tied to any anchor/query positions, arbitrary geometry representations can be encoded. Hence, this design choice makes the model mesh independent. Additionally, this improves performance, at the cost of additional training runtime for the geometry encoder.

**Micro design**. Subsequently, we aim to improve speed and accuracy. For automotive CFD simulations, we aim to predict surface and volume variables where we use an equal amount of surface and volume tokens. However, these two modalities differ in the way they are calculated in a classical simulation where surface variables have boundary condition $u = 0$. Therefore, we split surface and volume positions into separate branches ("+ split surface/volume branch"). This halves the number of tokens of each individual branch, resulting in faster training/inference speed and decreased performance, particularly for surface variables.

To reintroduce cross-branch interactions, we swap the keys and values of the two branches in every other block ("+ cross-branch interactions") as visualized in Figure 4. Therefore, surface anchors/queries can retrieve information from volume anchors, and volume anchors/queries can retrieve information from surface anchors, enabling information flow between both branches at no additional cost.

Finally, we introduce branch-specific decoders by using different weights for the last $L$ Transformer blocks ("+ branch-specific decoders"). Conceptually, this focuses on learning the physical simulation in the first $K$ blocks (as the weights are shared between surface and volume) while additionally allocating weights/compute to branch-specific decoding.

## 4.4 Benchmarking AB-UPT against other neural surrogate models

Table 4: Relative L2 errors (in %) of surface pressure $p_s$, volume velocity $u$ and volume vorticity $\omega$. For each model, we indicate if the model has the neural field property and whether they are mesh independent. Lower percentage values indicate better performance in terms of L2 error. We provide the results for ShapeNet-Car, AhmedML, and DrivAerML. AB-UPT outperforms other neural surrogate models, often by quite a margin.

| | ShapeNet-Car | | AhmedML | | | DrivAerML | | | Neural field | Mesh independent |
|---|---|---|---|---|---|---|---|---|---|---|
| | $p_s$ | $u$ | $p_s$ | $u$ | $\omega$ | $p_s$ | $u$ | $\omega$ | | |
| PointNet | 12.09 | 3.05 | 8.02 | 5.44 | 66.04 | 23.63 | 28.13 | 1747.7 | ✗ | ✗ |
| GRAPH U-NET | 10.33 | 2.49 | 6.46 | 4.15 | 53.66 | 16.13 | 17.98 | 540.6 | ✗ | ✗ |
| GINO | 13.28 | 2.53 | 7.90 | 6.23 | 71.81 | 13.03 | 40.58 | 131.7 | ✓ | ✓ |
| LNO | 9.05 | 2.29 | 12.95 | 7.59 | 72.49 | 20.51 | 23.27 | 493.8 | ✓ | ✓ |
| UPT | 6.41 | 1.49 | 4.25 | 2.73 | 15.03 | 7.44 | 8.74 | 90.2 | ✓ | ✓ |
| OFormer | 7.05 | 1.61 | 4.12 | 3.63 | 15.06 | 4.48 | 6.64 | 71.2 | ✓ | ✓ |
| Transolver | 6.46 | 1.62 | 3.45 | 2.05 | 8.22 | 4.81 | 6.78 | 38.4 | ✗ | ✗ |
| Transformer | 4.86 | 1.17 | 3.41 | 2.09 | 6.76 | 4.35 | 6.21 | 47.9 | ✗ | ✗ |
| **AB-UPT** | **4.81** | **1.16** | **3.01** | **1.90** | **6.52** | **3.82** | **5.93** | **35.1** | ✓ | ✓ |

In this section, we compare the performance of AB-UPT against established neural-surrogate models: Table 4 presents the relative L2 error for surface pressure, volume velocity, and vorticity. We compare AB-UPT against the following models: PointNet (Qi et al., 2017a), Graph U-Net (Gao & Ji, 2019), GINO (Li et al., 2023b), LNO (Wang & Wang, 2024), UPT (Alkin et al., 2024a), OFormer (Li et al., 2023a), Transolver (Wu et al., 2024), Transformer (Vaswani et al., 2017). We do not consider Transolver++ as a baseline due to reproducibility issues, which we explain in Appendix C.4 and B.7. These comparisons are conducted across three diverse datasets: ShapeNet-Car (Umetani & Bickel, 2018), AhmedML (Ashton et al., 2024a), and DrivAerML (Ashton et al., 2024b). A comparison on DrivAerNet++ (Elrefaie et al., 2024b) is conducted in Appendix B.1.

**Experimental setup**. For ShapeNet-Car, we use the same dataset split as in (Wu et al., 2024), with 789 simulation samples for training and 100 for testing; for AhmedML and DrivAerML, we use a random split of 400 training and 50 test samples, and the remaining samples (50 for AhmedML and 34 for DrivAerML) are used for validation. For a full comparison against baselines, we use 4K/4K, 16K/16K, 16K/16K surface/volume points for ShapeNet-Car, AhmedML, and DrivAerML, respectively. When computing metrics, we generate predictions using this number of inputs and repeat the process as many times until the full point cloud is processed. We then compute the evaluation metrics on the resulting predictions.

The 16K/16K surface/volume points for AhmedML and DrivAerML are motivated by the insights of Figure 5, showing that 16K obtains good results at relatively low compute costs. Evaluation procedure and evaluation metrics are detailed in Appendix C. Appendix C also presents the training hyperparameters and the number of model parameters for each model. Additionally, Appendix B.6 provides a statement on the fairness of our experimental setup in comparison to (Wu et al., 2024). Comprehensive results, including wall shear stress and volume pressure, are reported in Appendix B.8. Additional results for varying the number of slices in Transolver is provided in Appendix B.5 and comparison against Erwin (Zhdanov et al., 2025) is conducted in Appendix B.4.

**Performance comparison**. Table 4 shows that AB-UPT consistently outperforms all other neural surrogate models across every evaluated metric and dataset, often by a substantial margin. Moreover, we observe that most baselines struggle with properly modeling the complex vorticity. AB-UPT, however, maintains strong

performance for both surface and volume metrics, highlighting its robustness. For most metrics, either a plain Transformer or Transolver is the second-best-performing model. However, due to the full self-attention mechanism of the Transformer, this model will not be able to scale to industry-standard simulation meshes (Transformer corresponds to the third row of Table 3, which shows roughly 100 times slower test speed than AB-UPT). Transolver, on the other hand, would require sequence parallelism over multiple GPUs to handle large simulation meshes (as has been explored in (Luo et al., 2025)). We can conclude that AB-UPT is the best performing model for neural-surrogate automotive CFD simulation, while also having favorable properties concerning problem scaling.

## 4.5 Aerodynamic drag and lift coefficients

Accurate and fast predictions of global quantities, such as drag and lift coefficients, are crucial for designing efficient aerodynamic geometries. We evaluate the quality of the surface-level predictions of our model by computing these integrated aerodynamic quantities by predicting pressure $\mathbf{p}$ and wallshearstress $\boldsymbol{\tau}$ for each of the 9 million surface mesh cells using 16 thousand anchor tokens (employing *chunked inference*). These predictions are used as $\boldsymbol{p}$ and $\boldsymbol{\tau}$ in Equation 2 to calculate the force acting on the car $\boldsymbol{F}$. We then plug in $\boldsymbol{F}$ into Equations 3 to obtain the drag coefficient $C_d$ and the lift coefficient $C_l$. Figure 6 shows prediction and reference values for drag and lift coefficients on DrivAerML. AB-UPT obtains accurate predictions as indicated by the high coefficient of determination ($R^2$) score. Additionally, such integrated quantities require high-resolution predictions (see Figure 1), which AB-UPT can effortlessly produce via anchor attention, despite being trained on only a small subset (e.g., 16 thousand) surface points.

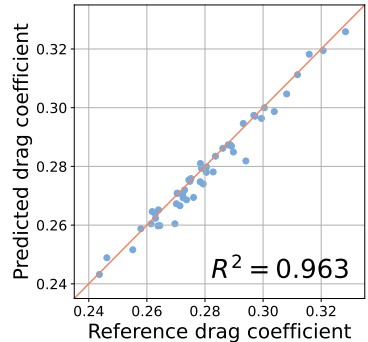
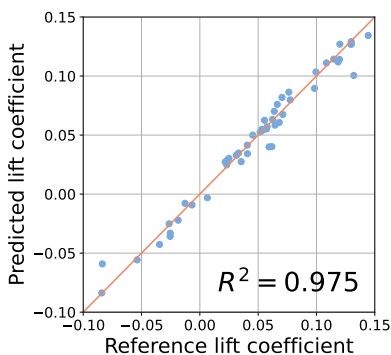

Figure 6: AB-UPT can accurately predict aerodynamic drag and lift coefficients of DrivAerML. To obtain accurate results, pressure and wall shear stress need to be predicted for millions of surface points (see Figure 1). AB-UPT can make accurate predictions for millions of surface points despite being trained with only 16 thousand surface points.

## 4.6 Training from CAD

A particularly intriguing property of AB-UPT is that it decouples query positions, anchor positions, and geometry encoding. This allows us to train models that can make predictions on geometry representations other than the meshed CFD geometry. To showcase this, we consider the following setting: we uniformly sample surface points from a CAD file and use these positions as inputs to the geometry encoder and as surface anchors. For volume anchors, we uniformly sample random positions in the domain. The model is then trained by using surface/volume positions from the CFD mesh as query points on which a loss is calculated. In this setting, no loss is calculated for anchor points since there is no corresponding CFD simulation results ground truth available for the anchor positions. As our model simulates physical dynamics only within the anchor tokens (query tokens do not influence anchor tokens as they only interact via cross-attention), this enables learning an accurate neural surrogate that operates on uniform meshes. Note that a numerical simulation would either not converge or produce suboptimal results on a uniform surface/volume mesh.

Table 5 compares the accuracy of such approaches. Sampling anchor positions from the CFD simulation mesh implicitly assigns more computational resources to highly resolved regions, which typically contain complicated dynamics. Uniformly sampling from the CAD geometry (surface) and random positions in the domain (volume) removes this bias and degrades performance slightly, as expected. However, this performance degradation is fairly small, particularly when considering that numerical simulations would produce completely unusable results. To further decrease this performance gap, anchor tokens could be sampled from a low-fidelity CFD mesh to approximate the anchor position distribution of the high-fidelity mesh, without requiring the expensive high-fidelity mesh generation process. We leave exploration of such approaches to future work.

Table 5: Relative L2 errors (in %) of neural simulation without using the CFD simulation mesh. AB-UPT can produce accurate results even in the absence of the CFD simulation mesh, which is typically costly to obtain. Numerical simulations (e. g., HRLES) would not converge or produce unrealistic results without the CFD simulation mesh. Note that HRLES with the CFD simulation mesh is used as ground truth (i.e., 0.00 error).

| | AhmedML | | | DrivAerML | | | Meshing | |
|---|---|---|---|---|---|---|---|---|
| | $p_s$ | $u$ | $\omega$ | $p_s$ | $u$ | $\omega$ | Surface mesh | Volume mesh |
| HRLES | 0.00 | 0.00 | 0.00 | 0.00 | 0.00 | 0.0 | CFD simulation mesh | CFD simulation mesh |
| HRLES | N.A. | N.A. | N.A. | N.A. | N.A. | N.A. | Geometry mesh (CAD) | Regular grid |
| AB-UPT | 3.01 | 1.90 | 6.52 | 3.82 | 5.93 | 35.1 | CFD simulation mesh | CFD simulation mesh |
| AB-UPT | 3.70 | 2.50 | 7.59 | 4.14 | 6.64 | 51.2 | Geometry mesh (CAD) | Regular grid |

## 4.7 Divergence-free vorticity

As discussed in Sections 3.2 and 3.3, the conditional field property of AB-UPT allows for defining vorticity field predictions that are divergence-free by design. We train the model in two stages: in the pretraining phase, we learn a model that predicts all surface and volume fields *except* the vorticity, using the same model configuration and hyperparameters as in the previous experiments; in the finetuning phase, we learn all fields *including* the derived vorticity field (cf. Section 3.3). Furthermore, we use float32 precision for finetuning, and we train with a smaller initial learning rate and fewer training steps: starting from $2e^{-5}$, the learning rate is decayed to $1e^{-6}$ via the cosine annealing schedule. As discussed in Section 3.3, after correcting for the scaling of the velocity field due to normalization, the predicted vorticity is already in *physics space*. To bring the predictions to a similar scale as other predicted fields, we transform both the predictions and targets as follows: first, divide by the field's standard deviation; then apply a sign-preserving square root to the vector's magnitude. This is equivalent to scaling the individual loss terms. All other fields are simply normalized as in all other experiments, subtracting the mean and dividing by the standard deviation (c. f. Appendix C.4.1).

Table 6 shows that the *divergence-free* vorticity formulation of AB-UPT matches the performance of the *direct-prediction* approach. However, while direct prediction results in non-zero divergence of the predicted vorticity field, the divergence-free formulation guarantees zero divergence by construction. Note that the most competitive baselines, such as Transolver and Transformer, do not have a (conditional) field formulation, and, thus, the local divergence operator is not defined for these models.

Table 6: Evaluation of AB-UPT with the divergence-free vorticity formulation (c. f. Section 3.3). For each dataset, the first 3 columns show the relative L2/L1 test errors (in %) of surface pressure $p_s$, volume velocity $u$, and volume vorticity $\omega$. The last column shows the mean absolute value of the divergence of the predicted vorticity field $\nabla \cdot \omega$ (in $m^{-1}s^{-1}$), where the average is taken over simulations and samples of the field.

| | AhmedML | | | | DrivAerML | | | |
|---|---|---|---|---|---|---|---|---|
| | $p_s$ | $u$ | $\omega$ | $\nabla \cdot \omega$ | $p_s$ | $u$ | $\omega$ | $\nabla \cdot \omega$ |
| AB-UPT (direct) | 3.26/1.69 | 1.98/1.09 | 5.23/4.02 | 13.69 | 3.98/3.11 | 5.64/5.49 | 33.0/15.74 | 586.7 |
| AB-UPT (divfree) | 3.24/1.69 | 2.01/1.10 | 5.57/4.30 | 0 | 3.82/3.09 | 5.58/5.40 | 31.39/16.16 | 0 |

## 4.8 Scaling AB-UPT

We investigate three axes of scaling (as visualized in Figure 2a): problem-scale, model-scale, and data-scale. In every study, one scale is varied, whereas the other two scales are fixed. By default, we use 16K/16K surface/volume anchor tokens (problem-scale), 12 blocks with a dimension of 192 (model-scale), and all 400 training simulations of DrivAerML (data-scale), which is the setting used throughout the previous sections.

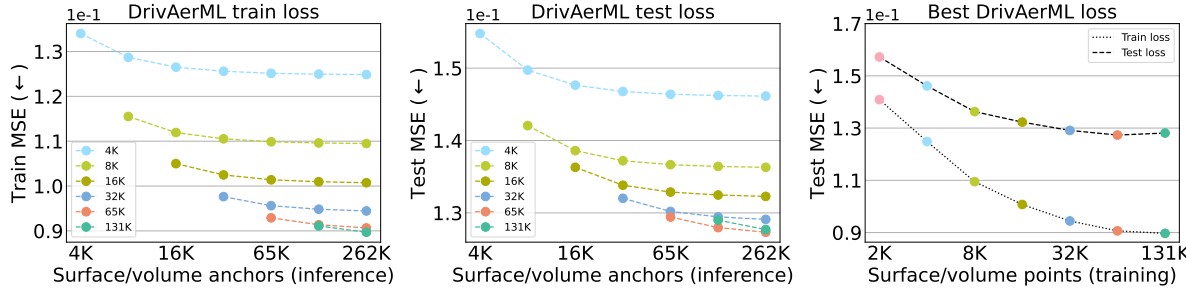

Figure 7: **Left and middle:** Scaling the number of anchor tokens *during inference only*. Models are trained with varying anchor tokens (as indicated by their starting point and label). These models are then evaluated with an increasing number of anchor tokens but without any retraining (as indicated by the x-axis). Different colors correspond to different numbers of anchor tokens used during training. An increased number of anchor tokens used in training yields performance improvements; an increased number of anchor tokens during inference yields similar accuracies. **Right:** Summary of the left and middle plot where the number of inference anchors is fixed at 262K (i.e., the best obtained loss) and the x-axis varies the number of training anchors instead. The performance saturation hints at the problem scale of the DrivAerML dataset, i.e., certain amounts of anchor tokens are enough to capture the intricacies of the dataset.

**Problem-scale**. We train AB-UPT models with various numbers of anchor tokens (4K to 131K) and additionally increase the number of anchor tokens further during inference only (up to 262K), *without any retraining*. By increasing the number of anchor tokens, we increase coverage of the 150M simulation mesh cells and observe saturating effects when increasing the number of anchor tokens beyond a certain point. Figure 7 shows results thereof. The left and middle plots show that AB-UPT benefits from an increased number of anchor tokens in inference time, but the best models are obtained by also training with more anchor tokens. As the number of anchor tokens increases, performance gains saturate (e.g., doubling training anchors from 4K to 8K decreases the test loss by 8%, doubling anchors from 32K to 64K decreases test loss by only 2%).

Interestingly enough, a model trained with 131K anchor tokens already shows signs of overfitting, where the train loss still decreases (is better than with 65K tokens) but test loss increases (is worse than with 65K tokens), as shown on the right side of Figure 7. This hints at the underlying problem-scale of DrivAerML, i.e., 65K tokens in training are sufficient to capture the underlying dynamics of the problem. Notably, this is three orders of magnitude less than the 150M mesh cells used in the classical CFD simulation.

**Model-scale**. We investigate two ways to scale model size: increasing the dimension of the model and increasing the depth of the model and present results in Figure 8. As expected, increasing model size improves the train loss. The test loss also increases until overfitting becomes a problem.

**Data-scale**. As obtaining more high-resolution CFD data is costly (roughly $2000 per simulation), we investigate scaling the dataset size by training on subsets of the full dataset. Figure 8 shows a clear trend where more data leads to better generalization performance.

**Compound-scaling**. We compare individually scaling axis against jointly scaling multiple axis at once. To this end, we train an AB-UPT model with 32K surface/volume anchors, model dim 384, and model depth 24.

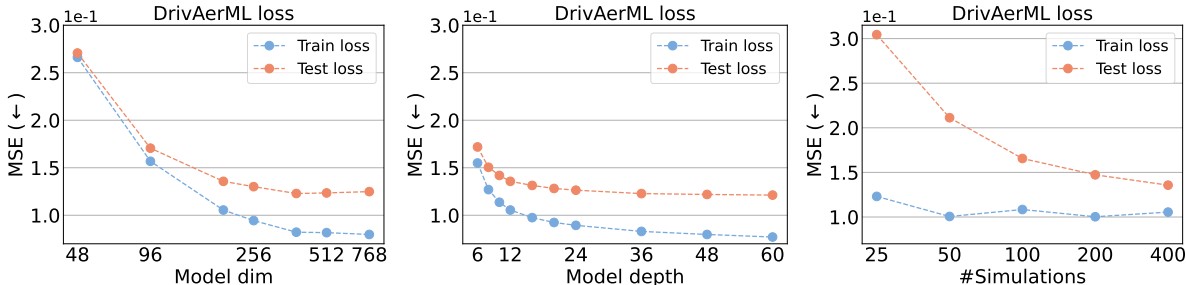

Figure 8: **Left and middle:** Model scaling by scaling either the dimension (using a depth of 12) or the depth (using a dimension of 192). **Right:** Scaling dataset size.

Table 7: Relative L2 errors (in %) of surface pressure $\boldsymbol{p}_s$, volume velocity $\boldsymbol{u}$, volume vorticity $\boldsymbol{\omega}$ on DrivAerML for different scaling variants and a compound scaled model. Further gains could most likely be obtained by increasing #Tokens and training duration for the compound scaled model (last row), which we leave for future exploration due to computational costs.

|  | Setting | | | DrivAerML | | | |
| --- | --- | --- | --- | --- | --- | --- | --- |
|  | #Tokens | Model dim | Model depth | Test loss | $\boldsymbol{p}_s$ | $\boldsymbol{u}$ | $\boldsymbol{\omega}$ |
| AB-UPT | 16K | 192 | 12 | 0.1314 | 3.82 | 5.93 | 35.1 |
| AB-UPT | **128K** | 192 | 12 | 0.1300 | 3.61 | 5.69 | 32.7 |
| AB-UPT | 16K | **384** | 12 | 0.1214 | 3.55 | 5.20 | 25.4 |
| AB-UPT | 16K | 192 | **60** | 0.1212 | 3.47 | 5.20 | 21.0 |
| AB-UPT | **32K** | **384** | **24** | **0.1169** | **3.38** | **4.87** | **18.8** |

## 5  Limitations and future work

**The bottleneck of quadratic complexity of self-attention for anchor tokens**. AB-UPT employs expressive self-attention among anchor tokens, which exhibits quadratic complexity. We show that on high-fidelity CFD simulations, a small set of anchor tokens is sufficient to obtain good results on currently available public datasets. However, using more anchor tokens can improve accuracy further, leading to (quadratically) increasing computational demands. Contrary, the computational demands of models with linear complexity would only increase linearly.

However, we did not manage to scale linear attention models to obtain higher accuracies than AB-UPT, even when using an order of magnitude more inputs (see Figure 5). Additionally, the trend of the loss curves in Figure 5 suggests that the benefit of scaling the number of inputs saturates. As there is still a sizable gap between the AB-UPT test loss and linear attention test loss (i.e., Transolver), it is unlikely that linear attention models would obtain better accuracies than AB-UPT, even when trained on a large number of inputs. Nevertheless, for an extremely large number of anchor tokens, compute demands would explode, and AB-UPT would need some kind of linear attention mechanisms. For example, ball attention (Zhdanov et al., 2025) is a promising attention mechanism with linear complexity that could be integrated into AB-UPT.

**Fast training through parallelization**. To keep training times low, a future direction would be to apply parallelization techniques such as sequence parallelism (Liu et al., 2023) or tensor parallelism (Shoeybi et al., 2019) to scale the number of anchor tokens to the next order of magnitude. Additionally, one could easily parallelize inference by dividing the query tokens among different GPUs, which would linearly decrease inference time.

**AB-UPT beyond automotive CFD**. AB-UPT is a general-purpose architecture, which we expect to be applicable to many use cases beyond automotive CFD. We chose automotive CFD as a problem setting due to existing publicly available high-fidelity datasets. However, as AB-UPT is a natural successor of

UPT, we expect it to perform well on many simulation types, as is the case for UPT (e.g., transient fluid simulations (Alkin et al., 2024a) or particle and particle-fluid multi-physics simulations (Alkin et al., 2024b)).

**Further improvements of AB-UPT and neural surrogates**. While AB-UPT shows strong results, there are many interesting directions for further improvement of AB-UPT and neural surrogate models in general. Instead of randomly sampling anchor tokens, the sampling could be informed by domain expertise (e.g., dense sampling at boundary regions), uncertainty estimates, or even employ some kind of learned sampling procedure (e.g., by learning a probability distribution over the domain or using learnable 3D vectors as anchor positions). Additionally, there are many avenues to improve the performance of neural surrogates in general, including physics-preserving data augmentation, jointly training on high- and low-fidelity simulations or other heterogeneous data sources, transfer learning to low data regimes, low-fidelity to high-fidelity transfer learning, and uncertainty quantification to warn users of potentially high error regions or out-of-distribution settings. We believe many of these techniques could improve AB-UPT and neural surrogates in general; however, these are beyond the scope of this work.

**High-fidelity CFD simulations**. The ever-lasting paradigm that a surrogate model can only achieve the accuracy of its underlying simulation is particularly valid in CFD. The chosen turbulence modeling approach (e.g., RANS, LES, HRLES) and the adopted meshing strategy critically determine the fidelity and accuracy of the simulation results. This is especially pronounced in external automotive aerodynamics, where capturing complex near-wall flow behavior holds significant challenges and remains a key area for methodological advancement. In this context, CFD approaches such as the Lattice Boltzmann Method (Chen & Doolen, 1998) are of growing interest due to their inherently parallelizable structure, which facilitates efficient deployment on GPU architectures. Accelerated simulation techniques enable the resolution of finer-scale flow structures and near-wall phenomena, thereby improving the physical realism of the computed flow fields. Enhancing the underlying simulation quality directly benefits the machine learning-based surrogate models, leading to higher accuracy, better generalization, and increased reliability.

**Fidelity vs variability trade-off**. In this paper, we mainly consider high-fidelity HRLES CFD simulations, which can be extremely expensive to obtain (see Table 2). Consequently, current publicly available datasets tend to either produce high-fidelity simulations with relatively few input variability or lower-fidelity simulations with more input variability. For example, DrivAerML (Ashton et al., 2024b) uses high-fidelity HRLES simulations with a single geometry class and static simulation parameters (e.g., Reynolds number), whereas DrivAerNet++ (Elrefaie et al., 2024b) uses lower-fidelity RANS simulations with multiple geometry classes, sacrificing simulation fidelity in favor of geometric variability (simulation parameters are also static for DrivAerNet++). Evaluating AB-UPT on datasets with more diverse geometries and flow conditions is an interesting future direction.

**Physics constraints via neural field formulation**. In Section 4.7, we show how formulating a neural surrogate model as a conditional neural field (e.g., via anchor attention) can be leveraged to adhere to physics constraints such as predicting divergence-free vorticity by estimating differential operators via finite-differences (or automatic differentiation). We show this capability on a comparatively simple use-case of predicting divergence-free vorticity. Future work could leverage the neural field formulation to impose additional physics constraints, such as mass-conservation of the velocity field. However, this is a non-trivial extension as the curl operator alone is not sufficient to model a mass-conserving velocity field due to potential additive components that are simultaneously divergence-free and vorticity-free (i.e, harmonic).

## 6    Conclusion

In this work, we answer the question of how to define and harness scale in the context of automotive CFD simulations. We introduce Anchored-Branched Universal Physics Transformers: a neural surrogate architecture for physics simulation, which we demonstrate on high-fidelity automotive CFD simulations with up to hundreds of millions of simulation mesh cells. We investigate the notion of problem scale, where the number of mesh cells required for industry-standard numeric CFD simulation is typically correlated with the complexity of physical dynamics. However, neural surrogate models learn a conceptually different solution to the numerical equations and can simulate complex dynamics in a heavily reduced resolution. This allows us to leverage expressive self-attention to build a powerful modeling backbone. However, practical use-cases often

necessitate a high-resolution prediction, e.g., to obtain accurate drag/lift coefficients. Therefore, we introduce *anchor attention*, which enables prediction at arbitrary resolution at linear complexity by leveraging a small set of *anchor tokens* as context to create predictions for arbitrary query positions.

AB-UPT outperforms strong neural surrogate benchmark models and has beneficial properties such as mesh independence, which allows neural simulation from CAD geometry alone, omitting the need for the costly creation of a CFD simulation mesh. Additionally, we make use of the neural field property of AB-UPT to formulate a divergence-free vorticity prediction to make predictions physically consistent.

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

# A    Model design ablations

## A.1    Model design on DrivAerML

Table 8: Impact of model design on DrivAerML. The design steps follow a similar pattern as in Table 3 where the impact of step-by-step model design choices are shown on AhmedML.

| | | L2 error ($\downarrow$) | | | Neural field | Mesh independent | Train speed |
|---|---|---|---|---|---|---|---|
| | | $p_s$ | $u$ | $\omega$ | | | |
| baseline | UPT | 12.67 | 16.65 | 10424 | ✓ | ✓ | 7.0 |
| macro design | + large decoder | 7.44 | 8.74 | 90.2 | ✓ | ✓ | 14.7 |
| | + decoder-only with self-attn | 4.35 | 6.21 | 47.9 | ✗ | ✗ | 14.9 |
| re-add properties | + anchor attention | 4.35 | 6.21 | 47.9 | ✓ | ✗ | 14.9 |
| | + decouple geometry encoding | 4.12 | 5.90 | 33.7 | ✓ | ✓ | 16.1 |
| micro design | + split surface/volume branch | 4.23 | 6.09 | 32.6 | ✓ | ✓ | 11.0 |
| | + cross-branch interactions | 4.24 | 5.97 | 35.8 | ✓ | ✓ | 11.0 |
| | + branch-specific decoders | 3.82 | 5.93 | 35.1 | ✓ | ✓ | 11.0 |

## A.2    Geometry branch design

We evaluate various design choices of the geometry branch in Table 9.

(a) Subtracting the position of the supernode before encoding the positions of the incoming nodes allows the model to develop local filters instead of learning the same filter for every location in the domain. Although performance is relatively similar, we choose relative positions to enforce learning local patterns in the message passing of the supernode pooling layer. Note that we concatenate the absolute position embedding after supernode pooling. This is conceptually similar to Vision Transformers (Dosovitskiy et al., 2021) where pixels are encoded into patches using local operations without absolute positions embedding. Afterwards, absolute positions are integrated into patch tokens.

(b) Increasing the number of geometry points, i.e., the resolution of the geometry input, does not significantly improve performance while increasing runtime.

(c) Connecting points to supernodes via a k-NN graph can obtain similar results to a radius graph. However, using a k-NN graph makes the model dependent on the number of geometry inputs as more points would reduce the distance to the furthest neighbor. The radius graph does not have this disadvantage as it is similar to Graph Neural Operators (Li et al., 2020).

(d) Even high-resolution geometries like the DrivAerML shapes can be captured with a relatively low number of geometry inputs and geometry tokens.

(e) Integrating the geometry information directly at the start of the model shows better results than integrating it later on or attending multiple times to the geometry tokens via multiple cross-attention blocks in the surface/volume branches.

(f) The geometry branch benefits from local aggregation, i.e., supernode pooling, followed by a global information exchange, i.e., a Transformer block. Adding more Transformer blocks is not necessary. Note that using neither supernode pooling nor Transformer blocks corresponds to not using the geometry branch at all where the cross-attention block is replaced by a self-attention block to keep the number of parameters of the surface/volume branch constant. Therefore, showing the benefits of using a geometry branch.

We note that the slight performance differences of (a-d) are negligible, and we choose designs/hyperparameters based on conceptual motivations (a, c) or lower runtimes (b, c).

Table 9: Geometry branch design study on DrivAerML. Test losses are multiplied by 10. Default settings

(a) **Position encoding in supernode pooling**

| | Test loss |
|---|---|
| Absolute in domain | 1.360 |
| Relative to supernode | **1.357** |

(b) **Number of geometry points**

| #Points | Test loss |
|---|---|
| 64K | **1.357** |
| 128K | **1.357** |

(c) **Connectivity**

| | Test loss |
|---|---|
| Radius graph (r=0.25) | **1.357** |
| k-NN graph (k=16) | 1.361 |
| k-NN graph (k=32) | 1.362 |

(d) **Number of geometry tokens**

| #Points | #Tokens | Test loss |
|---|---|---|
| 64K | 16K | 1.357 |
| 64K | 32K | **1.355** |
| 128K | 32K | 1.360 |

(e) **Location of cross-attention**

| Location | Test loss |
|---|---|
| Start | **1.357** |
| Mid | 1.425 |
| End | 1.395 |
| Start + mid | 1.390 |
| Start + end | 1.360 |

(f) **Number of Transformer blocks**

| Supernode pooling | #Blocks | Test loss |
|---|---|---|
| ✗ | 0 | 1.405 |
| ✓ | 0 | 1.370 |
| ✓ | 1 | **1.357** |
| ✓ | 2 | 1.366 |
| ✓ | 4 | 1.364 |

## A.3 Training without query tokens

As anchor attention shares its weights for the anchors and queries, one would ideally train a model without employing a loss on the queries, as propagating query tokens during training imposes a runtime overhead. However, there is a conceptual difference between how anchors interact with each other and how queries interact with anchors. Namely, anchor tokens attend to themselves and can influence the representation of other anchor tokens. While this is conceptually very similar to how query tokens are propagated, it imposes a train-test discrepancy if no query tokens are used during training. We investigate the influence of this discrepancy by training models where the loss is calculated on (i) only anchor tokens (ii) only query tokens (iii) anchor and query tokens. We then evaluate these models using a variable number of query tokens to investigate the generalization capabilities of anchor attention when trained with a loss on the anchor tokens. Table 10 shows that the train-test discrepancy from (i) leads to a slight decrease in performance, which is not present in (ii) or (iii). We use (i) throughout the main paper due to its computational efficiency.

Table 10: The train-test discrepancy of training with only anchor tokens (i), leads to a slight increase in loss when query tokens are used in inference. Including queries during training (ii) and (iii) leads to almost the same loss on anchors and queries. Training with a loss on anchors and queries (iii) obtains the best performance. Relative increase denotes the increase from the anchor loss to the query loss. We use DrivAerML with 16K anchors/queries for this study. Train speed denotes the runtime for 100 updates with batchsize 1.

| | Training with loss on | | Loss when evaluated on | | | |
|---|---|---|---|---|---|---|
| | Anchors | Queries | Anchors ($\downarrow$) | Queries ($\downarrow$) | Relative increase ($\downarrow$) | Train speed |
| (i) | ✓ | ✗ | 0.13574 | 0.13624 | $+0.37\%$ | 11.5s |
| (ii) | ✗ | ✓ | 0.13634 | 0.13632 | $-\mathbf{0.01}\%$ | 19.4s |
| (iii) | ✓ | ✓ | 0.13462 | **0.13468** | $+0.04\%$ | 19.4s |

Adding increasingly many query tokens adds computational costs and poses a trade-off. Figure 9 visualizes this trade-off, where an increasing number of queries improves the performance of the predictions from the query tokens. When employing a combination of anchor and query losses (iii), fewer query tokens can be used.

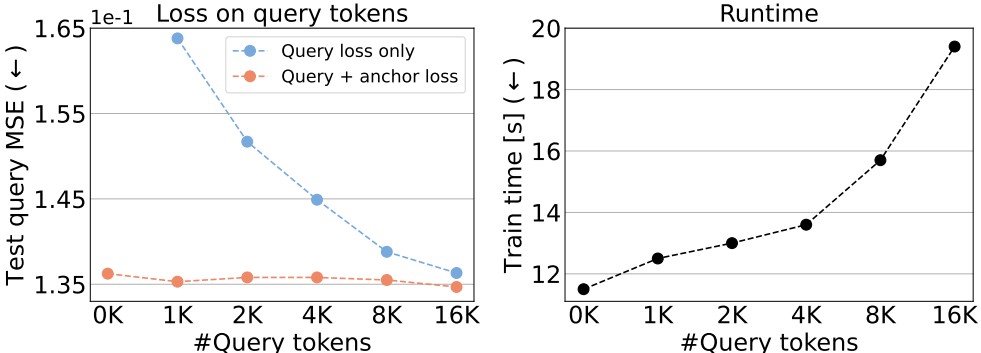

Figure 9: Performance of predictions from query tokens on DrivAerML when trained with various number of query tokens. The number of anchor tokens is fixed at 16K. The values at 16K query tokens correspond to (ii) and (iii) of Table 10 and the value at 0K corresponds to (i). Train time is measured as runtime for 100 updates with batchsize 1.

## A.4    Variance of anchor selection

We evaluate the variance in predictive performance originating from the random sampling process of the anchor tokens. To evaluate this, we change the evaluation protocol of Section 4.4 (as described in Section C.5.3) slightly. Instead of propagating only a chunk of 16K anchor tokens (without any query tokens), then concatenating all chunks of a sample and calculating the relative L2 error on the concatenated chunks, we instead measure performance on query tokens, where we use the concatenation of all chunks as query positions. This results in multiple predictions for the exact same locations, where the only difference is the selection of the anchor points, where mean and standard deviation are calculated per sample and then averaged over the whole test split of the dataset. Table 11 shows that AB-UPT is fairly robust against the random anchor selection process.

Table 11: Relative L2 errors (in %) of surface pressure $p_s$, volume velocity $u$, volume vorticity $\omega$, wall shear stress $\tau$, and volume pressure $p_v$ on the DrivAerML dataset. Lower percentage values indicate better performance in terms of L2 error. We evaluate the mean and standard deviation of a single seed of an AB-UPT model from Section 4.4 by fixing query locations and varying the anchor tokens.

|  | $p_s$ | $u$ | $\omega$ | $\tau$ | $p_v$ |
|---|---|---|---|---|---|
| Variable anchors (from Section 4.4) | 3.82 | 5.93 | 35.1 | 7.29 | 6.08 |
| Static queries with variable anchors | 3.81±0.06 | 5.98±0.06 | 37.0±4.0 | 7.31±0.09 | 6.18±0.11 |

## A.5    Anchor attention versus k-NN interpolation

Anchor attention updates the representation of a query token based on the anchor tokens, without influencing them. Additionally, we sample random points in space, so each query token most likely has a relatively close-by anchor token. To validate the effectiveness of anchor attention, we compare our approach against a baseline of a k-NN interpolation (Qi et al., 2017b), where we interpolate the anchor tokens after the full forward pass to query locations via a weighted sum of the 3 nearest neighbors of each query location, where the weights are based on the distance to each neighbor.

Table 12 shows that anchor attention, although only trained on a reduced resolution (e. g., trained with 16K anchor points), can produce high-fidelity predictions also for query points (e. g.., up to 140 million points). It is significantly better than a simple k-NN interpolation as the model learned non-linear interactions between anchor points during training, which it can transfer "zero-shot"-like to query points, even though the query points do not influence the anchor points. This underlines the effectiveness of our approach and confirms that our model learns more sophisticated prediction schemes than linear interpolation.

Table 12: Analysis of the effectiveness of anchor attention vs a simple k-NN interpolation. Anchor attention can learn more complex interactions than a simple interpolation. Values denote test MSE losses on AhmedML.

|  | AhmedML | | |
|---|---|---|---|
|  | $p_s$ | $u$ | $\omega$ |
| k-NN interpolation | 0.00703 | 0.06641 | 0.5748 |
| Anchor attention | 0.00276 | 0.00657 | 0.0397 |

### A.6 Mixed-precision training

We train all AB-UPT in fp16 mixed precision as this makes training and inference much faster. Previous works found instabilities when training with fp16 (Alkin et al., 2024a), which we also experienced in early development cycles. By keeping positional embeddings in fp32 precision and adding Rotary Positional Embeddings (RoPE) (Su et al., 2024) (also in fp32 precision), we stabilized training while preserving predictive accuracy and the large speedups of fp16 mixed precision training. To highlight this, we train AB-UPT in full fp32 precision on DrivAerML. Table 13 shows no significant predictive accuracy gain of fp32 training at vastly increased training times.

Due to the vast speedup of fp16 mixed precision training, we also train baselines in mixed precision whenever possible, where we also keep positional embeddings in fp32. For more detail,s see Appendix C.5.

Table 13: Relative L2 errors (in %) of surface pressure $p_s$, volume velocity $u$, volume vorticity $\omega$, wall shear stress $\tau$, and volume pressure $p_v$ on DrivAerML for different training precisions. Training time is measured with 500 epochs on a single NVIDIA H100 GPU.

|  | $p_s$ | $u$ | $\omega$ | $\tau$ | $p_v$ | Training time |
|---|---|---|---|---|---|---|
| AB-UPT (full fp32) | 3.78 | 5.94 | 36.1 | 7.25 | 6.07 | 33.6h |
| AB-UPT (mixed-fp16) | 3.82 | 5.93 | 35.1 | 7.29 | 6.08 | **6.9h** |

## B  Extended benchmark results

### B.1  DrivAerNet++ evaluation

We also evaluated AB-UPT on the DrivAerNet++ dataset (Elrefaie et al., 2024a;b), an open-source collection of over 8,000 automotive aerodynamic simulations with extensive geometric variations. While this dataset contains significantly more geometric variations than DrivAerML (Ashton et al., 2024b), a key limitation is its reliance on a lower-fidelity turbulence model, as the simulations were carried out using RANS. We compare AB-UPT against all baselines and use the same evaluation setup as in the main paper (see Appendix C.5.3). All models were trained to predict all fields at the same time as in our experiments in Section 4.4, except that volume vorticity is missing from this dataset. For AB-UPT, super node pooling used a radius of 1.0, the number of supernodes and the number of anchor points in the surface and volume branch was set to 16384 each, and, similarly to the main experiments (cf. Appendix C.3), the model uses 12 Transformer blocks, where the last 4 self-attention blocks do not share parameters. Furthermore, we used a learning rate of $1 \times 10^{-4}$ for AB-UPT and $5 \times 10^{-5}$ for the baselines, and we trained all models for 35 epochs. Surface pressure $p_s$, surface friction $\tau$, volume pressure $p_v$, and volume velocity $u$ are reported in Table 14. While AB-UPT outperforms

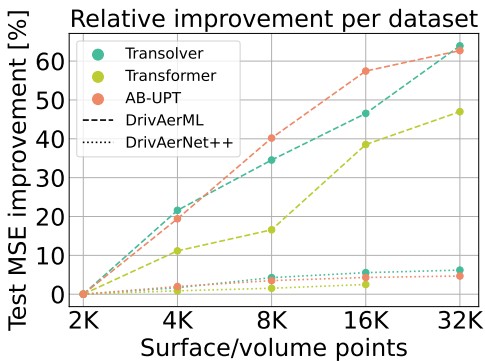

Figure 10: Relative improvement (in %) of the test error (MSE) when increasing the model capacity via the number of surface and volume points. Test MSE improvement denotes the test MSE at the respective number of surface/volume points divided by the obtained test MSE at 2K points. Results are shown for Transolver, Transformer, and AB-UPT on both DrivAerNet++ and DrivAerML. Since DrivAerNet++ does not have vorticity, we exclude it from DrivAerML as well for better comparison. Increasing the model's computational capacity leads to significant performance gains on DrivAerML, but has much smaller performance gains on DrivAerNet++.

Table 14: Relative L2 errors (in %) of surface pressure $\boldsymbol{p}_s$, wall shear stress $\boldsymbol{\tau}$, volume pressure $\boldsymbol{p}_v$, and volume velocity $\boldsymbol{u}$ on DrivAerNet++.

|  | $\boldsymbol{p}_s$ | $\boldsymbol{\tau}$ | $\boldsymbol{p}_v$ | $\boldsymbol{u}$ |
|---|---|---|---|---|
| PointNet | 21.62 | 31.76 | 20.07 | 16.75 |
| GRAPH U-NET | 18.68 | 27.42 | 16.13 | 13.62 |
| GINO | 15.45 | 23.74 | 16.24 | 13.80 |
| LNO | 21.87 | 31.93 | 18.00 | 15.68 |
| UPT | 14.31 | 22.16 | 12.55 | 11.06 |
| OFormer | 19.13 | 29.06 | 17.69 | 15.20 |
| Transolver | 13.97 | 21.73 | 12.44 | 11.06 |
| Transformer | 13.86 | 21.59 | 12.42 | 11.00 |
| **AB-UPT** | **13.79** | **21.49** | **12.34** | **10.94** |

all baselines, we notice that Transolver and Transformer follow much closer in performance compared to experiments on ShapeNet-Car, AhmedML, and DrivAerML in Table 4.

Furthermore, it is evident that all surrogate models achieve worse performance in terms of relative L2 error on DrivAerNet++ than they do on DrivAerML or ShapeNet-Car. We attempted to train better models by, e.g., increasing training duration, increasing model size, or increasing the number of (anchor) points per model. In these experiments, we found that the training loss steadily declines where AB-UPT obtains significantly lower train losses than Transolver and a vanilla Transformer, while test performance (measured by relative L2 errors) does not improve with any of the three considered models. We also observed that increasing model capacity leads to significantly less performance gains on DrivAerNet++ compared to DrivAerML (see Figure 10). This suggests that model capacity is not the limiting factor, but there could be significant irreducible noise in the dataset resulting from the lower-fidelity simulation method. Determining the primary cause of this performance discrepancy was not investigated further, but it would be a valuable direction for future research.

### B.2 Benchmarking AB-UPT against TripNet on DrivAerNet++

We compare AB-UPT against the recently proposed TripNet (Chen et al., 2025) on the DrivAerNet++ dataset (Elrefaie et al., 2024a;b). As there is no publicly available implementation of TripNet and the input

representation and pre-processing pipeline is completely different from our setup, we did not attempt to implement this baseline and instead compare directly to the results reported in Chen et al. (2025). We trained AB-UPT with the train/validation/test split provided by Elrefaie et al. (2024b) and report median evaluation metrics over 5 training runs. For evaluating the vector-valued fields, we follow the experimental setup of Chen et al. (2025) and report the relative L2 error of the wall shear stress *magnitude* $|\boldsymbol{\tau}|$ and velocity *magnitude* $|\boldsymbol{u}|$. The results are summarized in in Table 15, and show that AB-UPT significantly outperforms TripNet on all metrics.

Table 15: Relative L2 errors (in %) of surface pressure $\boldsymbol{p}_s$, surface friction wall shear stress magnitude $|\boldsymbol{\tau}|$, and volume velocity magnitude $|\boldsymbol{u}|$, and the velocity components in the respective directions. Lower percentage values indicate better performance in terms of L2 error. We provide the results for DrivAerNet++. AB-UPT outperforms TripNet, often by quite a margin. The results for TripNet are taken from (Chen et al., 2025)*.

| | Surface | | Volume | | | |
|---|---|---|---|---|---|---|
| | $\boldsymbol{p}_s$ | $|\boldsymbol{\tau}|$ | $|\boldsymbol{u}|$ | $\boldsymbol{u}_x$ | $\boldsymbol{u}_y$ | $\boldsymbol{u}_z$ |
| TripNet* | 20.05 | 22.07 | 10.39 | 10.71 | 35.34 | 36.39 |
| AB-UPT | **13.79** | **18.26** | **8.48** | **8.63** | **32.18** | 30.98 |

## B.3 Benchmarking AB-UPT against DoMINO

Table 16 presents a direct comparison between AB-UPT and DoMINO (Ranade et al., 2025). DoMINO is originally trained and tested using a specific data split, where 20% of the test samples are out-of-distribution, based on the range of drag force values.

We trained AB-UPT with the same train/test split as provided in (Ranade et al., 2025), and report median evaluation metrics over 5 training runs. All other hyperparameters for AB-UPT remained consistent with those reported in Appendix C. The numbers for DoMINO are taken directly from (Ranade et al., 2025) and we compare to the same metrics.

Based on Table 16, we can conclude that AB-UPT outperforms DoMINO with a sufficient margin when both models are trained and evaluated on the same data splits.

Table 16: Relative L2 errors (in %) of surface pressure $\boldsymbol{p}_s$, wall shear stress $\boldsymbol{\tau}$ per $x, y, z$, dimension, volume velocity $\boldsymbol{u}$ per $x, y, z$ dimension, and volume pressure $\boldsymbol{p}_v$. Lower percentage values indicate better performance in terms of L2 error. We provide the results for DrivAerML. AB-UPT outperforms DoMINO, often by quite a margin. The results for DoMINO are taken from (Ranade et al., 2025)*.

| | Surface | | | | Volume | | | |
|---|---|---|---|---|---|---|---|---|
| | $\boldsymbol{p}_s$ | $\boldsymbol{\tau}_x$ | $\boldsymbol{\tau}_y$ | $\boldsymbol{\tau}_z$ | $\boldsymbol{p}_v$ | $\boldsymbol{u}_x$ | $\boldsymbol{u}_y$ | $\boldsymbol{u}_z$ |
| DoMINO* | 15.05 | 21.24 | 30.2 | 33.59 | 21.93 | 23.97 | 50.25 | 45.67 |
| AB-UPT | **3.76** | **5.35** | **3.65** | **3.63** | **6.29** | **4.43** | **3.04** | **2.61** |

## B.4 Benchmarking AB-UPT against Erwin

Erwin (Zhdanov et al., 2025) is a promising linear complexity Transformer variant that employs self-attention within local regions, so-called *balls*, with efficient memory structures to enable computational efficiency on arbitrary pointclouds. This changes the runtime complexity of the attention from $\mathcal{O}(N^2)$ to $\mathcal{O}((N/B) * B^2)$ where $N$ is the number of inputs, $N/B$ is the number of balls and $B$ is the ball size, i.e., the number of tokens within a single ball. If $B = N$, it reduces to full self-attention.

As similar local attention variants designed for regular grids (Liu et al., 2021) have been very successful employed in neural simulation (e.g., weather modeling (Bodnar et al., 2025)), we consider ball attention a promising direction with one of the best runtime-accuracy tradeoff among linear complexity Transformers. Combinations of AB-UPT and ball attention are interesting future directions, where anchor attention could

Table 17: Benchmarking our adaptation of Erwin to jointly model surface and volume data on ShapeNet-Car. Relative L2 errors (in %) of surface pressure $p_s$, and volume velocity $u$. Erwin (reimpl.) with ball size 8192 would be roughly equivalent to the Transformer row.

| | Ball size | $p_s$ | $u$ |
|---|---|---|---|
| *Linear complexity Transformers* | | | |
| LNO | - | 9.05 | 2.29 |
| OFormer | - | 7.05 | 1.60 |
| Transolver | - | 6.46 | 1.62 |
| Erwin (reimpl.) | 512 | 5.46 | 1.37 |
| Erwin (reimpl.) | 1024 | 5.38 | 1.37 |
| Erwin (reimpl.) | 2048 | 5.35 | 1.27 |
| Erwin (reimpl.) | 4096 | **5.31** | **1.25** |
| *Quadratic complexity Transformers* | | | |
| Transformer | 8192 | 4.86 | 1.17 |
| AB-UPT | 4096 | **4.81** | **1.16** |

be extended to support local ball structures. Similarly the cross-attention components of AB-UPT could be adapted to cross ball attention.

However, as shown in the main paper (e.g., Figure 5), training full quadratic self-attention models is perfectly feasible even on the problem-scale of DrivAerML employing >100M mesh cells for its HRLES simulations. Consequently, it is unlikely that the local attention obtains better accuracy than full self-attention but if runtime is a requirement, local attention could speedup the model while largely preserving accuracy. In the following, we aim to provide insights into this trade-off by adapting ball attention to our joint surface/volume variable modeling setting. This adaption, termed *Erwin (reimpl.)*, uses an (i) isotropic architecture, i.e., there are no up/downsampling operations in the model, (ii) does not use local attention biases and (iii) uses two single layer message passing modules for surface/volume encoding respectively, instead of three layers of message passing as in the original architecture. (i) Is used to have a roughly amount of parameter count to AB-UPT. (ii) Speeds up computations and keeps the vanilla self-attention computation instead of biasing it. (iii) Aligns message passing complexity with that of supernode pooling where multi-layer message passing would become extremely costly with large number of inputs. After encoding surface and volume points, we concatenate them and process them with 12 ball attention blocks with a dimension of 192 (same setting as for the other transformer-based models). We do not add rotational embeddings (RoPE) (Su et al., 2024) to ball attention blocks as the original implementation[4] does not support it, which could yield further performance improvements. Also hierarchical designs could yield further improvements, which we do not consider as there is a large design space for hierarchical designs and they are hard to directly comparable in terms of FLOPS/parameter counts.

We first compare on ShapeNet-Car where we can easily study variable ball size counts. Table 17 shows that Erwin is a very promising linear complexity Transformer variant performing best on ShapeNet-Car among the considered ones in this paper. However, quadratic complexity Transformers obtain better accuracies.

Next, we compare our reimplementation of Erwin on DrivAerML in the default setting considered in the main paper (16K/16K surface/volume points) and show results in Figure 11. We observe that ball attention imposes an overhead, most likely from memory restructuring operations to structure tokens for efficient attention computations. This overhead becomes negligible with more inputs, and ball attention becomes faster than (branched) self-attention. As there is a significant accuracy gap between AB-UPT trained on 16K/16K points and Erwin (reimpl.) trained on 16K/16K points, we hypothesize that more sophisticated training protocols are necessary to train ball tree attention models to sufficient quality (e.g., hierarchical architectures, different ball size to input token ratios, etc.). As the design space thereof is large and fair comparisons to AB-UPT become more and more difficult, we leave exploration of this direction to future work.

---

[4]https://github.com/maxxxzdn/erwin

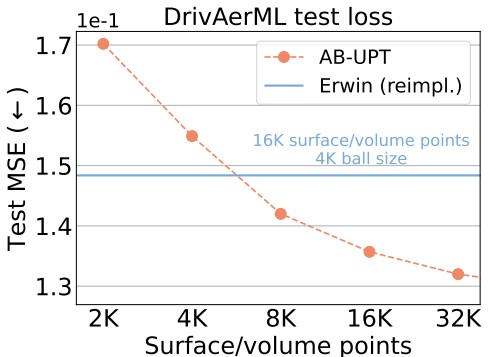 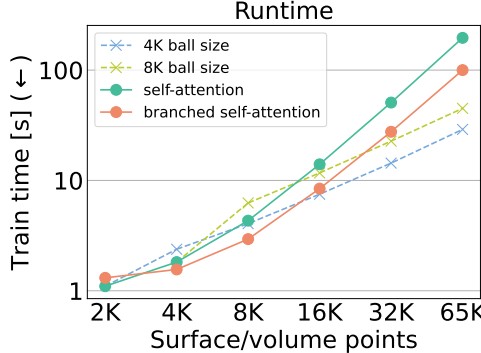

Figure 11: **Left:** Test loss of Erwin (reimpl.) trained on DrivAerML. **Right:** Runtime comparison of plain self-attention and ball attention blocks. Train time is measured in seconds per 100 updates with batch size 1 with 12 blocks of dimension 192.

Finally, we compare AB-UPT against the original Erwin architecture (including three-layer message passing and distance-based attention biases) in the setting used in Zhdanov et al. (2025). Table 18 confirms that our reimplementation obtains sensible results and that AB-UPT also performs best with this protocol.

Table 18: Comparing AB-UPT and Erwin (reimpl.) against Erwin on ShapeNet-Car surface pressure only prediction (no volume data is used). We use a similar protocol to Erwin (300 epochs with early stopping, 700/189 train/test split from (Alkin et al., 2024a), median of 5 runs). Results for Erwin are taken from (Zhdanov et al., 2025). Note that Erwin (reimpl.) uses 12 blocks of dimension 192 and consequently has more parameters than Erwin-M. Ball size is 256.

|  | $p_s$ MSE |
|---|---|
| Erwin-S | 15.85 |
| Erwin-M | 15.43 |
| Erwin (reimpl.) | 15.17 |
| AB-UPT | **13.25** |

### B.5 Extended comparison against Transolver

In Transolver (Wu et al., 2024), the quadratic attention is limited to a subset of tokens by *slicing* the input domain into a reduced domain (i.e., physics tokens). Throughout the experiments in the main paper, we use 512 slices. In this section, we investigate the impact of this hyperparameter in terms of performance (test loss), memory consumption, and runtime. Figure 12 shows that increasing the number of slices slightly improves test loss at the cost of additional memory and runtime requirements. As the slicing mechanism is implemented via slow and memory-heavy `torch.einsum` operations (following the original implementation[5]), it requires much more memory than AB-UPT, which leverages the constant memory complexity of FlashAttention (Dao et al., 2022) (which results in constant memory complexity for the attention layer, but the overall memory complexity is still linear). For larger problem sizes (e.g., DrivAerML), the memory bottleneck becomes significant and requires memory-saving techniques such as activation checkpointing to fit onto a single GPU, which in turn increases runtime. Note that we improved speed and memory consumption of the original Transolver implementation by using FlashAttention for the self-attention between slice tokens and employing mixed-precision training. Further optimizations (e.g., specialized hardware implementations of the transolver attention) are certainly possible but are beyond the scope of this work. Note that we also do not optimize AB-UPT beyond the use of FlashAttention[6] with mixed-precision training. Also AB-UPT could be optimized further, e.g., by employing FlashAttention-3 (Shah et al., 2024).

---

[5] https://github.com/thuml/Transolver/blob/main/Car-Design-ShapeNetCar/models/Transolver.py
[6] `torch.nn.functional.scaled_dot_product_attention` using PyTorch 2.4.1 with CUDA 12.4

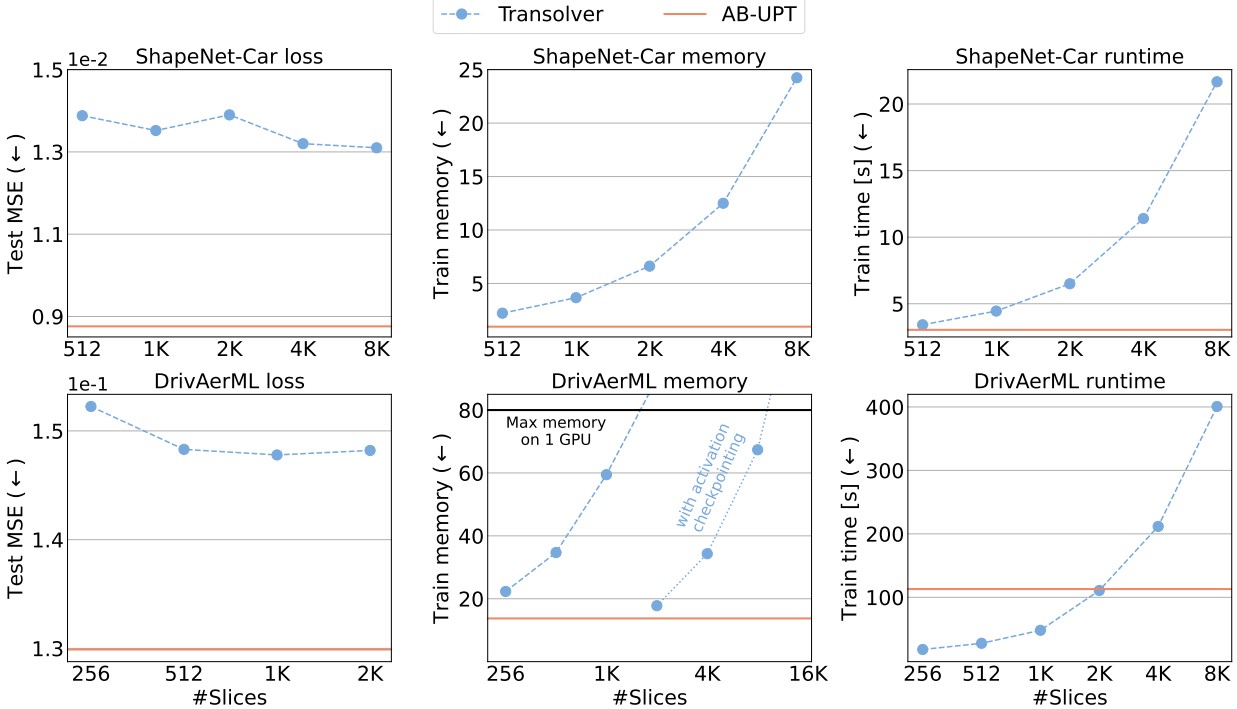

Figure 12: Performance (left), memory consumption (center) and runtime (right) on ShapeNet-Car (top) and DrivAerML (bottom) of transolver with variable number of slices (i.e., latent tokens). Increasing the number of slices slightly improves performance, but does not come close to AB-UPT. Additionally, increasing the number of slices greatly increases memory and compute demands. The number of surface/volume points for this study is 4K/4K for ShapeNet-Car and 65K/65K for DrivAerML.

## B.6    Transolver and baseline reproducibility

Table 19: Comparison of our baseline model evaluated on ShapeNet-Car, run in the same experimental set-up as in the Transolver paper. Evaluating median and best performance over 5 training runs with different seeds for $p_s$ and $u$, alongside the results reported in the Transolver paper. The results for AB-UPT are from Table 4 (*).

| Model | Median $p_s$ | Median $u$ | Best $p_s$ | Best $u$ | Transolver paper $p_s$ | Transolver paper $u$ |
|---|---|---|---|---|---|---|
| PointNet | 12.33 | 4.51 | 10.84 | 3.56 | 11.04 | 4.94 |
| GRAPH U-NET | 9.87 | 3.97 | 9.66 | 3.88 | 11.02 | 4.71 |
| GINO | 9.32 | 2.69 | 9.21 | 2.64 | 8.10 | 3.83 |
| LNO | 7.71 | 2.26 | 7.51 | 2.14 | - | - |
| UPT | 8.77 | 2.20 | 6.90 | 2.00 | - | - |
| OFormer | 6.58 | 1.70 | 6.52 | 1.66 | - | - |
| Transolver | 7.21 | 2.00 | 6.97 | 1.90 | 7.45 | 2.07 |
| Transformer | 5.75 | 1.51 | 5.61 | 1.34 | - | - |
| AB-UPT* | 4.81 | 1.18 | - | - | - | - |

In this work, we train and evaluate AB-UPT and all the baseline models in our unified experimental framework. Some of our baselines are taken from the Neural-Solver-Library (Wu et al., 2024). To validate the correctness of our experimental framework and baseline implementations, we rerun all the ShapeNet-Car experiments

with the experimental setup (i.e., data loading, training hyper-parameters, model parameters) as reported in Wu et al. (2024) (including the same input features: SDF, input coordinates, and normal vectors) in our own framework. We report the median/best pressure ($\boldsymbol{p}_s$) and velocity ($\boldsymbol{u}$) values over five training runs with different seeds, and where possible, we compare the values with the evaluation metrics reported in Wu et al. (2024). For each Transformer-based model, we use 8 Transformer blocks, with 8 attention heads, and a hidden dimensionality of 256, similar to Transolver.

As shown in Table 19, our best-performing models either match or surpass the reported results from the Transolver paper. Notably, the median results for the Transolver baseline are especially close to those reported in the original publication. Our AB-UPT model (median results from Table 4, without the additional SDF and normal vectors) still outperforms all baselines with a sufficient margin. All other baselines not reported in the Transolver paper performed as anticipated. The goal of these experiments is not to achieve better (baseline) performance but to validate the fairness and correctness of our experimental setup. Based on these results, we conclude that our experimental framework is robust and fair when compared to the work by Wu et al. (2024). Any differences observed with the values in Table 4 are attributable to consistently applied changes for all models within our experimental setup. While our results in general align with those reported in Wu et al. (2024), slight variations are still there due to randomness in model training and unavoidable minor implementation differences.

### B.7 Reproducibility of Transolver++

Table 20: Reproducing Transolver++ on ShapeNet-Car. We started with Transolver, and step-by-step turned the implementation into Transolver++ by: (i) removing the over-parameterization of the slicing network, (ii) using Gumbel-Softmax instead of the softmax function, and (iii) adding the adaptive temperature parameter. We report relative L2 errors for surface pressure $\boldsymbol{p}_s$ and volume velocity $\boldsymbol{u}$ for a single training run.

|  |  | L2 error ($\downarrow$) | |
|---|---|---|---|
|  |  | $\boldsymbol{p}_s$ | $\boldsymbol{u}$ |
| Transolver |  | 6.48 | 1.57 |
|  | + reparameterization | 6.72 | 1.63 |
|  | + Gumbel-Softmax | 7.28 | 1.63 |
| Transolver++ | + adaptive temperature | 7.57 | 1.76 |

Transolver++ (Luo et al., 2025) is the parallelizable, more efficient, and accurate successor to Transolver (Wu et al., 2024). As Transolver++ had no publicly available code during the development of AB-UPT, we attempted to implement it on our own. Our goal was to include it in our baseline benchmark experiments, as shown in Table 4. Turing Transolver into Transolver++ involves three steps (excluding the parallelization)

   (i) Removing the over-parameterization of the slicing (+ reparameterization);

  (ii) Use a Gumbel-Softmax instead of a softmax (+ Gumbel-Softmax);

 (iii) Use adaptive temperature parameters (+ adaptive temperature).

In Table 20, we report the L2 error for the surface pressure and volume velocity for Transolver and the results after applying each individual step sequentially. Unfortunately, all the additions that define Transolver++ consistently led to worse evaluation performance.

Additionally, we train Transolver++ models using the publicly available implementation[7] in the same setting as used in Section 4.4. Table 21 shows that Transolver++ performs consistently worse than vanilla Transolver and AB-UPT on all considered datasets.

As both our own independent reimplementation and the results with the public implementation show worse results than the vanilla Transolver, we decided to exclude Transolver++ as a benchmark model in this work.

---

[7]https://github.com/thuml/Transolver_plus

Table 21: Relative L2 errors (in %) of Transolver++ models on all considered datasets. Transolver++ shows consistently worse results than Transolver and AB-UPT. Training settings for Transolver and Transolver++ are identical except for the attention mechanism and follow the setup used for Section 4.4.

| | ShapeNet-Car | | AhmedML | | | DrivAerML | | |
|---|---|---|---|---|---|---|---|---|
| | $p_s$ | $u$ | $p_s$ | $u$ | $\omega$ | $p_s$ | $u$ | $\omega$ |
| Transolver | 6.46 | 1.62 | 3.45 | 2.05 | 8.22 | 4.81 | 6.78 | 38.4 |
| Transolver++ | 7.75 | 1.74 | 4.08 | 2.35 | 9.24 | 5.26 | 7.16 | 40.3 |
| **AB-UPT** | **4.81** | **1.16** | **3.01** | **1.90** | **6.52** | **3.82** | **5.93** | **35.1** |

## B.8 Full results: benchmarking AB-UPT against other neural surrogate models

We provide the full benchmark results of all available surface and volume fields in Table 22.

Table 22: Relative L2 errors (in %) of surface pressure $p_s$, volume velocity $u$, volume vorticity $\omega$, wall shear stress $\tau$, and volume pressure $p_v$. Lower percentage values indicate better performance. We provide the results for ShapeNet-Car, AhmedML, and DrivAerML. AB-UPT outperforms other neural surrogate models, often by quite a margin. For ShapeNet-Car, only surface pressure and volume velocity are available.

| | ShapeNet-Car | | AhmedML | | | | | DrivAerML | | | | |
|---|---|---|---|---|---|---|---|---|---|---|---|---|
| | $p_s$ | $u$ | $p_s$ | $u$ | $\omega$ | $\tau$ | $p_v$ | $p_s$ | $u$ | $\omega$ | $\tau$ | $p_v$ |
| PointNet | 12.09 | 3.05 | 8.02 | 5.44 | 66.04 | 10.09 | 6.13 | 23.63 | 28.13 | 1747.70 | 41.85 | 31.24 |
| GRAPH U-NET | 10.33 | 2.49 | 6.46 | 4.15 | 53.66 | 7.29 | 5.18 | 16.13 | 17.98 | 540.67 | 27.84 | 20.51 |
| GINO | 13.28 | 2.53 | 7.90 | 6.23 | 71.81 | 8.18 | 8.10 | 13.03 | 40.58 | 113.67 | 21.71 | 44.90 |
| LNO | 9.05 | 2.29 | 12.95 | 7.59 | 72.49 | 11.50 | 8.48 | 20.51 | 23.27 | 493.77 | 36.44 | 27.01 |
| UPT | 6.41 | 1.49 | 4.25 | 2.73 | 15.03 | 5.80 | 3.10 | 7.44 | 8.74 | 90.2 | 12.93 | 10.05 |
| OFormer | 7.05 | 1.61 | 4.12 | 3.63 | 15.06 | 4.60 | 4.08 | 4.85 | 6.64 | 71.17 | 8.92 | 7.11 |
| Transolver | 6.46 | 1.62 | 3.45 | 2.05 | 8.22 | 4.00 | 2.16 | 4.81 | 6.78 | 38.4 | 8.95 | 7.74 |
| Transformer | 4.86 | 1.17 | 3.41 | 2.09 | 6.76 | 4.03 | 2.16 | 4.35 | 6.21 | 47.9 | 8.26 | 6.27 |
| **AB-UPT** | **4.81** | **1.16** | **3.01** | **1.90** | **6.52** | **3.88** | **1.98** | **3.82** | **5.93** | **35.1** | **7.29** | **6.08** |

## B.9 Velocity streamlines visualization

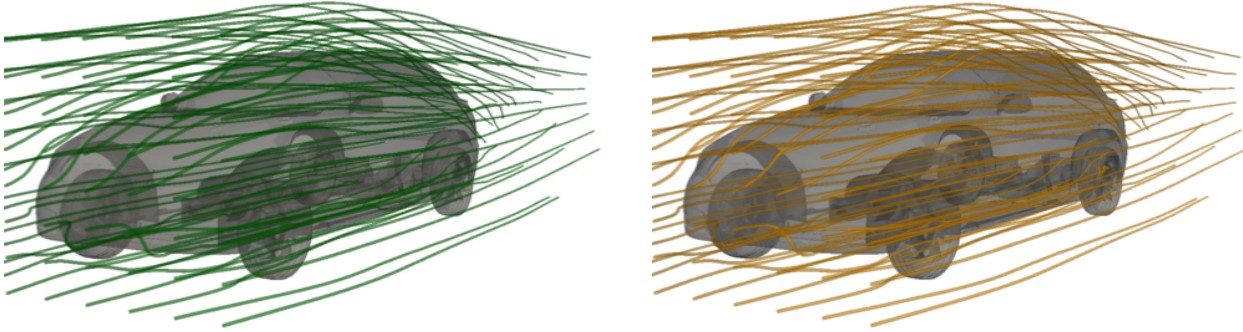

Figure 13: Velocity streamlines inferred on a DrivAerML sample. We plot the streamlines using the ground truth velocity values (left) and using velocity values predicted by AB-UPT (right).

## C  Experimental setup

### C.1  Datasets

All datasets consider exclusively time-averaged quantities.

**ShapeNet-Car**. The ShapeNet-Car dataset, a subset of the ShapeNet dataset (Chang et al., 2015) as introduced by Umetani & Bickel (2018), specifically focuses on the car-labeled data points. Each simulation in this dataset includes a surface mesh with 3,682 points and a corresponding volume mesh with 28,504 points. Following (Alkin et al., 2024a), we remove outlier points that do not belong to the surface mesh (which results in a total of 3,586 points per surface mesh). As model input, we only consider the points on the surface mesh (i.e., without using additional signed distance function values or surface normals). Our models are then trained to predict pressure values on the surface mesh and velocity values within the volume. We adopt the same training and testing split as in Wu et al. (2024), which reserves `param0` of the dataset for testing. This results in 789 samples for training and 100 samples for testing.

**AhmedML**. AhmedML (Ashton et al., 2024a) is an open-source dataset that provides high-fidelity CFD simulation results for 500 geometric variations of the Ahmed car body, a widely studied bluff body in automotive aerodynamics. The dataset includes hybrid RANS-LES simulations performed using OpenFOAM, capturing essential flow physics such as pressure-induced separation and 3D vortical structures. Each mesh in the dataset contains approximately 20 million cells. Since the dataset does not provide a predefined split for training, validation, and testing, we randomly divide the data into 400 train, 50 validation, and 50 test samples. Our models are then trained to predict pressure values and wall shear stress on the surface mesh and velocity, pressure, and vorticity values within the volume.

**DrivAerML**. The DrivAerML (Ashton et al., 2024b) dataset is designed for machine learning for high-fidelity automotive aerodynamic simulation. The dataset contains 500 parametrically morphed variants of DrivAer vehicles, aiming to address the challenge of the availability of open-source data for large-scale (in terms of the size of the simulation mesh) computational fluid dynamics (CFD) simulations in automotive aerodynamics. DrivAerML runs the CFD simulations on approximately 140 million volumetric mesh grids with Hybrid RANS-LES (Spalart et al., 2006; Chaouat, 2017; Heinz, 2020; Ashton et al., 2022), which is the highest-fidelity CFD approach used by the automotive industry (Hupertz et al., 2022; Ashton et al., 2024b). Each mesh in the dataset contains approximately 8.8 million surface points, with pressure and wall shear stress values on the surface and velocity, pressure, and vorticity values in the volume. When computing the drag and lift coefficient with AB-UPT, all 8.8 million points on the surface mesh are used. Since the dataset does not provide a predefined split for training, validation, and testing, we randomly divide the data into 400 train, 50 validation 34 validation (minus the 16 hidden samples), and 50 test samples. The dataset does not contain results for 16 DrivAer vehicles, which we assign to the validation set, resulting in 34 effective validation samples.

### C.2  Evaluation metrics

In this work, we mainly report the relative L2 error. The relative L2 error between a predicted vector of a point cloud $\hat{\mathbf{Y}} \in \mathbb{R}^{N \times d_{out}}$ and its ground truth counterpart $\mathbf{Y} \in \mathbb{R}^{N \times d_{out}}$, where $N$ is the number of points and $d_{out}$ is the dimensionality of the target output vector, is defined as

$$\mathrm{L2}_{\mathrm{rel}}(\mathbf{Y}, \hat{\mathbf{Y}}) = \frac{\left\|\hat{\mathbf{Y}} - \mathbf{Y}\right\|_2}{\|\mathbf{Y}\|_2} = \frac{\sqrt{\sum_{n=1}^{N} \sum_{d=1}^{d_{out}} (\hat{\mathbf{Y}}_{n,d} - \mathbf{Y}_{n,d})^2}}{\sqrt{\sum_{n=1}^{N} \sum_{d=1}^{d_{out}} \mathbf{Y}_{n,d}^2}}.$$

Our dataset consists of a set of $M$ target vectors in point clouds $\mathcal{Y} = \{\mathbf{Y}^{(1)}, \mathbf{Y}^{(2)}, \dots \mathbf{Y}^{(M)}\}$ and corresponding predictions $\hat{\mathcal{Y}} = \{\hat{\mathbf{Y}}^{(1)}, \hat{\mathbf{Y}}^{(2)}, \dots \hat{\mathbf{Y}}^{(M)}\}$ where the average relative L2 error is defined as:

$$\mathrm{L2}_{\mathrm{rel}}(\mathcal{Y}, \hat{\mathcal{Y}}) = \frac{1}{M} \sum_{m=1}^{M} \mathrm{L2}_{\mathrm{rel}}(\mathcal{Y}^{(m)}, \hat{\mathcal{Y}}^{(m)}).$$

During training, the target values are normalized to approximately mean 0 and standard deviation 1, where we additionally apply a log scaling to the vorticity values. The evaluation metrics are computed on the unnormalized targets and predictions.

### C.3 AB-UPT

In the following subsection, we use the a/b/c notation to indicate hyperparameters for ShapeNet-Car/AhmedML/DrivAerML, respectively. We use a radius of 9/0.25/0.25 for the supernode pooling in the geometry branch. The geometry branch consists of a supernode pooling layer, followed by a single Transformer block with self-attention. The number of supernodes in the geometry branch is 3586/16384/16384, which we also use as the number of surface anchor points in the surface branch. The number of volume anchor points is 4096/16384/16384. When comparing against baselines, we train without any query tokens, i.e., we create predictions for each anchor token and calculate the loss for those. In other settings (e.g., when training with the CAD geometry), we also use query tokens during training, where we set the number of query tokens to be equal to the number of anchor tokens.

We choose the architecture such that surface/volume tokens traverse 12 Transformer blocks. The first block in each branch is a cross-attention block to the output of the geometry branch. The next 8/8/4 blocks consist of interleaved cross-attention and self-attention blocks, where cross-attention blocks exchange information between branches by using tokens of the other branch as keys and values. The self-attention blocks operate only within their branch. The last 3/3/7 blocks are all self-attention blocks, where the last 2/2/6 self-attention blocks do not share parameters. Afterwards, a linear layer decodes the token representation into 4 surface variables (1 pressure dimension, 3 wall shear stress dimensions) and 7 volume variables (1 pressure dimension, 3 velocity dimensions, 3 vorticity dimensions).

We use a standard ViT (Dosovitskiy et al., 2021)-style pre-norm Transformer blocks where we additionally add Rotary Position Embeddings (RoPE) (Su et al., 2024) to all blocks. For the message passing of the supernode pooling layer, we use the positions relative to the supernode and the magnitude of the distance as input to the message passing. This is then followed by embedding the position of the supernode, concatenating it with the output of the message passing, and down-projecting it to the original dimension in order to integrate absolute positions after the message passing. This is in contrast to Alkin et al. (2024a), where the concatenation of the positions of the two connected nodes is used directly for message passing.

We train models for 500 epochs using batch size 1, LION optimizer (Chen et al., 2023) with peak learning rate 5e-5, weight decay 5e-2, a linear warmup for 5% of the training duration, followed by cosine decay with end learning rate of 1e-6. We train in float16 mixed precision and employ a mean squared error. Training takes roughly 3/7/7 hours on a single NVIDIA H100 GPU and occupies 4GB of GPU memory. Similar to Wu et al. (2024), we choose batch size 1 because it obtains the best accuracy.

### C.4 Baseline models

**GINO**. The Geometry Informed Neural Operator (GINO) (Li et al., 2023b) is a neural operator with a regularly structured latent space that learns a solution operator of large-scale partial differential equations. It exhibits, as AB-UPT, a decoupling of its geometry encoder and the field-based decoder. To allow for an efficient application of the Fourier Neural Operator (FNO) (Li et al., 2021), GINO transforms an irregular grid into a regular latent grid. In particular, it starts with employing a Graph Neural Operator (GNO) to map the irregular point cloud input to a regularly structured cubic latent grid. This structured latent space is then processed by the FNO. As a last stage, a second GNO block is employed as a field decoder to get query-point-based predictions in the original irregular point cloud space (e.g., on the surface manifold of a car geometry). To ensure a fair comparison with other baselines, all of which rely solely on the point cloud geometry, we also remove the SDF input feature for GINO. For our GINO implementation (similar to the one in (Alkin et al., 2024a)), we use a latent resolution of $64^3$, 16 Fourier modes, and a message passing radius of 10. For GINO, we train two models: one for surface and one for volume predictions. GINO maps both the surface and volume mesh to a regular grid. However, the volume mesh is much larger than the surface mesh, and hence, the geometry of the car is only represented by a small fraction of the entire regular grid, which leads to a degradation of the surface predictions.

**Graph U-Net**. Graph U-Net (Gao & Ji, 2019) is a specialized U-net network architecture designed for irregular grids, specifically graphs. To enable the U-Net architecture to work effectively with irregular graph structures, the Graph U-Net introduces two critical operations: a Graph Pooling layer that acts as the downsampling mechanism and a Graph Unpooling layer that restores the graph to a larger size. These two components allow the Graph U-Net to maintain the U-shaped encoder-decoder structure and skip connections. We use the public implementation from the Neural-Solver-Library (Wu et al., 2024).[8] To construct the neighborhood structure of the graph, we use K-nearest-neighbors, with $k = 20$.

**LNO**. The Latent Neural Operator (LNO) is a transformer-based neural operator that operates in a low-dimensional latent space, rather than the high-dimensional input/geometry space. LNO consists of an embedding layer that maps the raw input features/function to high-dimensional latent representations. A physics-cross-attention block that maps the input embedding to a low-dimensional number of latent tokens. Followed by several Transformer blocks that process these latent tokens. Finally, a decoder that maps the processed latent tokens back to the input space, predicting the PDE solution. We use the standard ViT-tiny configuration with 12 Transformer blocks, a channel size of 192, 3 self-attention heads, and an up-projection ratio of 4. Our implementation is based on (Wang & Wang, 2024), but adapted for the 3D aerodynamics.[9]

**PointNet**. PointNet (Qi et al., 2017a) is a point-based baseline model that processes the input point cloud at both local and global levels. Each point in the input point cloud is first mapped through an MLP. Next, a max-pooling operation is applied to these point features to obtain a global feature representation that captures the structure of the entire point cloud. Then, each point feature vector is concatenated with the global feature vector to give global context to each point representation. Finally, the combined representation is mapped through another MLP to predict the output signal for each point on the input surface mesh. We use the public implementation from the Neural-Solver-Library (Wu et al., 2024), where we tune $n_{\text{hidden}}$ per dataset.[10]

**OFormer**. The OFormer (Li et al., 2023a) is a transformer-based encoder-decoder architecture which leverages linear attention mechanism (Cao, 2021) to deal with high-dimensional problems/functions. The raw input features are first embedded with RoPE positional information into tokens. Next, several Transformer blocks are applied. Finally, a decoder (i.e., cross-attention + MLP) maps output queries into the PDE solution. Our implementation is based on (Li et al., 2023a), but adapted for the 3D aerodynamics.[11] For our implementation, we use Galerkin attention. We use the standard ViT-tiny configuration with 12 Transformer blocks, a channel size of 192, 3 self-attention heads, and an up-projection ratio of 4.

**Transolver**. Transolver (Wu et al., 2024) is a transformer-based baseline model and, at the time of writing, the state-of-the-art on ShapeNet-Car. It introduces the Physics-Attention mechanism, where each layer in the Transolver model takes a finite discrete point cloud representation of an input geometry as input, and maps each point to a learnable slice (also referred to as physics tokens). Points with similar physical properties are mapped to the same slice. First, each surface mesh point is mapped to slice weights, which indicate the degree to which each point belongs to a slice. Next, the slice weights are used to aggregate point features into physics-aware tokens. Multi-head self-attention is applied to these physics-aware tokens, rather than directly to the input points, which reduces the computational cost of the self-attention layer. Finally, after the self-attention layer, the physics-aware tokens are transformed back to mesh input points by deslicing. Afterward, a feed-forward layer is applied to the individual input point representations.

We reimplement Transolver according to the official implementation of (Wu et al., 2024)[12] and using a standard ViT-tiny configuration with 12 Transolver blocks, a channel size of 192, 3 self-attention heads, and an up-projection ratio of 4 for the feed-forward layers.. We use 512 slices in the Transolver attention, which we chose after varying the number of slices, where we did not obtain significant improvements for larger numbers of slices.

---

[8]https://github.com/thuml/Neural-Solver-Library/blob/main/models/Graph_UNet.py
[9]https://github.com/L-I-M-I-T/LatentNeuralOperator/blob/main/LNO-PyTorch/ForwardProblem/module/model.py
[10]https://github.com/thuml/Neural-Solver-Library/blob/main/models/PointNet.py
[11]https://github.com/BaratiLab/OFormer
[12]https://github.com/thuml/Transolver/blob/main/Car-Design-ShapeNetCar/models/Transolver.py

**Transolver++**. We investigated training Transolver++ (Luo et al., 2025) on the considered datasets. However, contrary to the results of the original paper, we found it to perform consistently worse than the vanilla Transolver (Wu et al., 2024). Therefore, we do not consider it in the experiments section. We refer to Appendix B.7 for the full reproduction attempts and experimental results.

**UPT**. The Universal Physics Transformer (UPT) (Alkin et al., 2024a) is a unified neural operator without a grid- or particle-based latent structure, that can be applied to a variety of spatio-temporal problems. Similar to GINO (Li et al., 2023b), UPT allows for querying the latent space at any point in the spatial-temporal domain. The input function, which is represented as a point cloud, is first mapped to a lower-dimensional (in terms of the number of input tokens) representation by a (message passing) supernode pooling layer. Next, a transformer-based encoder stack maps the supernode representations into a compressed latent representation. We do not use perceiver pooling as the increased compute efficiency of further compressing the input comes at the cost of accuracy, where we value accuracy over efficiency in our work. To query the latent space at any location, cross-attention blocks are used. To give approximately equal compute to all baseline models, we use 12 cross-attention blocks instead of 1.

### C.4.1 Data pre-processing

**Input coordinates**. For most baselines, the input coordinates are normalized with the mean and standard deviation of the input domain of the train set. For AB-UPT, GINO, OFormer, and UPT, however, scale the input coordinates within a range of $[0, 1000]$.

**Output targets**. The surface/volume, velocity, and vorticity are all normalized with mean and standard deviation. Next to that, we use a log-scale for the vorticity.

**Input size**. For Shapenet-Car, we train our models with 3586 points from the surface mesh and 4096 points in the volume. For both AhmedML and DrivAerML, we train our models with 16384 surface and volume points. To make the comparison with baseline models possible in Section 4.4, we chunk the evaluation meshes into chunks of 16384 for both the surface and volume.

### C.5 Training details

### C.5.1 General configurations

For each baseline model, we used the AdamW (Loshchilov & Hutter, 2019) optimizer, with the exception of AB-UPT/UPT/Transformer, where we use LION (Chen et al., 2023). We tuned the learning rate of models trained with AdamW by sweeping over $\{1e^{-3}, 5e^{-3}, 2e^{-3}, 5e^{-3}, 1e^{-4}, 5e^{-5}, 2e^{-5}\}$ to achieve optimal performance per baseline. Whenever LION is used, we use $5e^{-5}$ as this value has performed consistently well. Training was conducted using either float16 or bfloat16 precision, depending on which yielded the best results. An exception was made for GINO, as it exhibited unstable behavior with float16/bfloat16 mixed precision. All models were trained for 500 epochs with a batch size of 1. We applied a weight decay of 0.05 and a gradient clipping value of 1. A cosine learning rate schedule was implemented, including a 5% warm-up phase and a final learning rate of $1e^{-6}$

For all Transformer-based models, we use the ViT-tiny configuration (Touvron et al., 2021). I.e., 12 blocks, 3 attention heads, and a hidden dimensionality of 192. This is to make sure that all our transformer-based models are in the same range of trainable parameters.

### C.5.2 Model specific hyper-parameters

Per model, we tuned some model-specific details:

**PointNet**. We tuned the expansion factor of the hidden dimensionality in a range of $\{16, 32, 64, 128\}$.

**LNO**. For LNO, we tuned the number of latent modes (i.e., the number of latent tokens) in a range of $\{128, 256, 512, 1024, 2056\}$.

**Graph-UNet**. To create the graph structure of the input meshes, we use k-nearest-neighbors with $k = 20$. We use an $n_{hidden}$ of 128.

**GINO**. For GINO, we use a grid size of 64, a max number of input neighbors of 10, and a radius graph of 5.

**UPT**. We use the same settings as for AB-UPT for radius (9 for ShapeNet-Car, 0.25 for AhmedML/DrivAerML) and number of supernodes (3586 for ShapeNet-Car, 16384 for AhmedML/DrivAerML).

Table 23: Training hyper-parameters and the number of trainable parameters for each model. Where a parameter varies across datasets, it's presented in the format ShapeNet-Car, AhmedML, DrivAerML; otherwise, a single value applies to all.

| Model | LR | Precision | Model specific hyper-params | Model Params |
|---|---|---|---|---|
| PointNet | $\{1e^{-3}, 5e^{-4}, 5e^{-4}\}$ | float16 | 64 | 3.6M |
| GRAPH U-NET | $1e^{-3}$ | {float16, bfloat16, float16} | - | 14.1M |
| GINO | $\{1e^{-4}, 2e^{-4}, 5e^{-4}\}$ | float32 | - | 15.6M |
| LNO | $\{5e^{-4}, 2e^{-4}, 1e^{-3}\}$ | {bfloat16, float16, bfloat16} | $\{2056, 256, 1024\}$ | 6.3M |
| UPT | $5e^{-5}$ | float16 | $\{3586, 16384, 16384\}$ | 11.0M |
| OFormer | $\{2e^{-4}, 2e^{-4}, 1e^{-3}\}$ | bfloat16 | - | 6.1M |
| Transolver | $1e^{-3}$ | bfloat16 | - | 5.5M |
| Transformer | $5e^{-5}$ | float16 | - | 5.5M |
| AB-UPT | $5e^{-5}$ | float16 | - | $\{6.7M, 6.7M, 8.8M\}$ |

### C.5.3 Evaluation variance reduction

When evaluating against baseline models (including AB-UPT), we use the following protocol:

(i) Each model is trained 5 times with different random seeds, where the final result is the median of the 5 models;
(ii) The final model weights of each run are an exponential moving average (EMA) of model weights from previous updates (EMA update factor = 0.9999);
(iii) Each test sample is divided into chunks to match the number of inputs used during training, where the number of volume cells is first subsampled to match the number of surface cells. All chunks of a sample are concatenated before calculating metrics.
(iv) Evaluation results are averaged over 10 evaluation runs (i.e., we run 10 times over the entire evaluation set).

These precautions aim to facilitate a fair comparison against baseline models, as we observed a high variance in experimental results due to the small number of train/test samples and optimal results being obtained when using a batch size of 1.

**Motivation**. (i) Reduces the impact of model initialization, dataset shuffling, and point sampling. (ii) Averages gradient noise from the low batch size, since we train with a batch size of 1 similar to (Wu et al., 2024). (iii) Is motivated by the fact that baseline models without a neural field decoder would require complicated multi-GPU and multi-node inference pipelines to evaluate million-scale meshes (e.g., AhmedML and DrivAerML). Additionally, using more inputs during testing than during training creates a train-test discrepancy, leading to performance drops if models are not designed for that. (iv) Aims to smooth out noise from the stochastic evaluation process; stochasticity is induced via (iii) and model-specific stochastic processes, such as choosing anchor tokens. For (i), we report the median model performance over different seeds to compensate for performance outliers.

Note that this protocol was designed with limitations of baseline models in mind. AB-UPT can decode meshes of any size on a single GPU, train with batch size > 1 on a single GPU (i.e., memory constraints are not the reason why we train with batch size 1), and decode more inputs during inference than in training.

