# OpenReview forum: "AB-UPT: Scaling Neural CFD Surrogates for High- Fidelity Automotive Aerodynamics Simulations via Anchored- Branched Universal Physics Transformers"
_TMLR — Accepted by TMLR_

### Review · Reviewer_b9ft · 2025-08-11

**Summary Of Contributions:**

Authors propose a novel architecture suitable for large-scale CFD applications.

The proposed architecture is a modification of Universal Physics Transformer that improve scalability by several mechanisms, including: (i) decoupling of geometry encoder from the rest of data, (ii) special form of attention mechanism (anchor attention) that scales linearly with the number of points, (iii) chunking inference strategy that allows to lower memory pressure at inference.

The architecture is evaluated on three benchmarks of increasing scale. The results show that the architecture perform better that the baselines both from the point of accuracy and the scalability.

**Audience:**

Yes

**Audience Explanation:**

Authors study the scalability of deep learning approaches to industrial CFD simulations. Given the practical stance of the contribution and its focus on realistic numerical simulations I believe the contribution will be of interest for many TMLR readers.

**Claims And Evidence:**

Yes

**Claims Explanation:**

Most of the claims are well supported.

However, I find several claims debatable:
1. "Enabling continuous field predictions from CAD geometries, eliminating costly mesh creation."
2. "Accurate and physically consistent vorticity predictions via divergence-free formulation."

The claims are not completely baseless, but in my view they are slightly misrepresent the contribution of the authors. More details are available in the section Requested Changes.

**Requested Changes:**

**Architecture design**

Authors get into considerable details describing the architecture design. However, several points are still unclear to me. Can the authors please clarify:

1. What are the inputs to the network? From equations (4) -- (7) it seems that only coordinates are involved. Are there any other inputs like initial velocity?
2. Is it correct that "Surface points" and "Volume points" on Figure 4 refers to the points where the user want to find some physical field, e.g., velocity or pressure?
3. In section 3.2 authors describe anchor attention. Anchor points are randomly sampled from query points. This seems to imply that query points can not be arbitrary selected. For example, if I only decide to use small number of query points (e.g., $3$) or query points that are not distributed well enough over computational domain, the method would not properly work. It is correct?
4. Another question on anchor attention. The whole approaches looks related to supernode pooling. Can the authors please discuss the difference between these two approaches?

   Is not it possible to replace it with supernode pooling? More specifically, we receive a sufficiently uniform mesh of query points, perform message passing to aggregate information from neighbours of supernodes, after that process the fixed number of supernodes and decode resulting physical field with cross attention with selected query points?
5. The selection of anchors should also influence the accuracy of predictions. What is a typical distribution of accuracy when different anchors are chosen?
6. It looks like the architecture is very close to UPT, and authors acknowledge this explicitly by providing appropriate reference. Still, can authors please compare UPT and AB-UPT on some simple problem? For example, suppose we want to predict (for all points in $[0, 1]^3$) temperature field $T(x, y, t)$ that evolves according to the standard heat equation. What would be the difference between UPT and AB-UPT in this case?

**Datasets and metrics**
Description of the datasets and metrics is available in Appendix B. From the available description it is not entirely clear which variables are predicted.

More specifically, authors consider NS equation (equation (1) on page 5), so physical variables available from numerical simulation should be spatiotemporal fields, e.g., $\boldsymbol{u}(\boldsymbol{x}, t)$. As I understand, all metrics authors report are for spatial fields $\boldsymbol{\widetilde{u}}(\boldsymbol{x})$. Given that my question is: what happens to time variable $t$? Do authors predict averaged velocity $\boldsymbol{\widetilde{u}}(\boldsymbol{x})=\frac{1}{\Delta t}\int_{t}^{t+\Delta t}d\tau \boldsymbol{u}(\boldsymbol{x}, \tau)$ or the velocity at some particular moment in time $\boldsymbol{\widetilde{u}}(\boldsymbol{x}) = \boldsymbol{u}(\boldsymbol{x}, T)$ or something else entirely?

**Claim on vorticity**

It seems to me that the claim on vorticity is a somewhat overemphasised.

In equation (11) on page 12 authors explain that they simply approximate vorticity by postprocessing velocity with finite difference approximation.

I have several objections on the significance of this contribution:
1. Vorticity linearly depend on velocity, so to recover the vorticity is a matter of postprocessing.
2. The procedure itself is straightforward and not novel. Physics-informed neural network are available that enforce divergence-free condition exactly, e.g., https://arxiv.org/abs/2210.01741

I suspect that the finite-difference approximation alone with no end-to-end training will not provide good results for vorticity prediction from velocity. However, I believe a tiny GNN can be trained on predicted velocities to recover good approximation to vorticity. Did the authors try this kind of postprocessing by training additional simple model?

Can the authors please comment on why they consider this part of their work as important contribution?

**Claim on CAD geometry predictions**

As I understand authors claim that CAD data can be used directly without the need to create a mesh. I have several questions on that claim:
1. The mesh is required to obtain a data from numerical simulation. Is it the case that at best the mesh can be ignored only at inference? if this is true, I suggest to specify this explicitly in the paper.
2. The claim that CAD data can be used seems to be problem-specific. Suppose I represent some terrain (mountains and valleys) with CAD file and is interested in wind flow in this environment. Alternatively, one may consider a wind flow in the city layout. The selection of points on the surface is straightforward and can be done as authors suggest. The selection of points in the volume does not seem to be straightforward and the proper selection resembles an initial stages of mesh construction. How the authors suggest to select points in the volume when geometry is more involved?

**Scaling**

In my view we see (Figure 8) quick saturation of performance measure when the size of the architecture or the size of the dataset are increased. What is the opinion of the authors on that? Do they expect that it will be possible to increase accuracy significantly (on a few orders) by increasing the size of the transformer and the size of the dataset?

---

> ### Author Response · Authors · 2025-09-03
> **Rebuttal to review of Paper5252 by Reviewer b9ft (1/4)**
>
> Thank you for your review. We tried to address all your comments by either providing more context in the rebuttal /or following all your suggestions by adding material to the paper.
> We will mainly reply to the **Requested Changes** comments for our rebuttal:
>
> ## Architecture design
>
> - *What are the inputs to the network? From equations (4) -- (7) it seems that only coordinates are involved. Are there any other inputs like initial velocity?*
>
> In this work, we only use coordinates as input to the network. However, as we also state in the paper (Section 3.1, paragraph "Multi-branch encoder block"), the input to the network is not limited to input coordinates, and it would be possible to include additional physics features in the input of the network too.
>
> - *Is it correct that "Surface points" and "Volume points" on Figure 4 refers to the points where the user want to find some physical field, e.g., velocity or pressure?*
>
> Yes, that is correct. We generally assume that anchors and queries are disjoint subsets of all positions, in which case the model predicts physical fields at all input positions. However, one could choose to omit the predictions from the anchor tokens and only use the query tokens for making predictions (as investigated in Appendix A.1 and required for Section 4.6). We added more clarification regarding this point in the caption of Figure 4
>
> - *In section 3.2 authors describe anchor attention. Anchor points are randomly sampled from query points. This seems to imply that query points can not be arbitrary selected. For example, if I only decide to use small number of query points (e.g.,) or query points that are not distributed well enough over computational domain, the method would not properly work. It is correct?*
>
> Anchor tokens are randomly sampled from the input tokens (not from the query tokens), and self-attention is carried out only between anchor tokens. Therefore, anchor attention gives the same result independent of how many query tokens are used as query tokens do not influence the anchor tokens. However, we do see how the standard formulation could suggest that there is a coupling between anchor and query tokens. Therefore, we added a sentence to explain that, for notational simplicity, we formulate anchor attention where anchors and queries come from the same set. However, this is not necessary as shown, e.g., in Section 4.6, where anchor tokens are sampled from the CAD geometry (surface anchor tokens) or from a regular grid (volume anchor tokens) and query tokens are obtained from the CFD mesh (both surface and volume query tokens).
>
> - *Another question on anchor attention. The whole approaches looks related to supernode pooling. Can the authors please discuss the difference between these two approaches?*
>
> Anchor attention and supernode pooling are both methods to change the resolution but are "inverse" to each other, where supernode pooling **decreases** spatial resolution by aggregating local information, and anchor attention uses the anchors as support points to **increase** spatial resolution by informing query tokens. Supernode pooling aggregates local information from a typically larger point cloud into supernodes, informing the subsequent layers about the geometry sampled at a higher resolution. For example, our default setting uses 65K points as input to the geometry branch, which are compressed into 16K geometry tokens. Contrary, the anchor tokens serve both as "neural simulation domain" and as support points for query tokens, enabling accurate high-resolution outputs.
>
> - *Is not it possible to replace it with supernode pooling? More specifically, we receive a sufficiently uniform mesh of query points, perform message passing to aggregate information from neighbours of supernodes, after that process the fixed number of supernodes and decode resulting physical field with cross attention with selected query points?*
>
> While it is possible to replace (or omit) the supernode pooling, e.g., by simply using more points to obtain higher resolution geometry information directly in the anchor tokens, we find that supernode pooling is an efficient and performant way to integrate high resolution geometry information without drastically increasing the runtime. For example, our default setting uses 65K inputs to the geometry branch and 16K surface anchors. If one were to use 65K surface anchors and no geometry branch, the model would be more than 4x slower due to the increased sequence length and quadratic cost of the attention between anchors.

---

> ### Author Response · Authors · 2025-09-03
> **Rebuttal to review of Paper5252 by Reviewer b9ft (2/4)**
>
> - *The selection of anchors should also influence the accuracy of predictions. What is a typical distribution of accuracy when different anchors are chosen?*
>
> Thanks for highlighting this interesting aspect. We added an evaluation by varying the anchor tokens with a fixed set of query tokens and calculating mean and standard deviation on the predictions as Appendix A.8. The results show that there is fairly little variance mostly around 0.1\% relative L2 error.
>
> | Method                                           | Surface pressure        | Velocity          | Vorticity | Wallshearstress | Volume pressure        |
> |--------------------------------------------------|----------------|----------------|-----------------------|---------------------|----------------|
> | Variable anchors (from Section~4.4) | 3.82           | 5.93           | 35.1                  | 7.29                | 6.08           |
> | Static queries with variable anchors             | $3.81 \pm 0.06$ | $5.98 \pm 0.06$ | $37.0 \pm 4.0$        | $7.31 \pm 0.09$     | $6.18 \pm 0.11$ |
>
> - *It looks like the architecture is very close to UPT, and authors acknowledge this explicitly by providing appropriate reference. Still, can authors please compare UPT and AB-UPT on some simple problem? For example, suppose we want to predict (for all points in ) temperature field that evolves according to the standard heat equation. What would be the difference between UPT and AB-UPT in this case?*
>
> While AB-UPT is derived by using UPT as a starting point (as outlined in Section 4.3), we introduced multiple advancements to improve performance on industry-scale CFD (AhmedML, DrivAerML) simulations, but also on heavily simplified small-scale simulations (ShapeNet-Car). As outlined in Table 4, AB-UPT also significantly outperforms UPT on ShapeNet-Car. If a problem is sufficiently simple, one could most likely use a plain UPT architecture to obtain comparable results to AB-UPT with faster runtimes, but this is not the focus of our work, and we would also argue that most use-cases are sufficiently complicated that increased model capacity benefits predictive accuracy. AB-UPT can model much more complicated structures and dynamics as shown throughout the paper, which would make AB-UPT the preferable choice for such use-cases.
>
> ## Datasets and metrics
>
> - *Description of the datasets and metrics is available in Appendix B. From the available description it is not entirely clear which variables are predicted. More specifically, authors consider NS equation (equation (1) on page 5), so physical variables available from numerical simulation should be spatiotemporal fields, e.g.,. As I understand, all metrics authors report are for spatial fields. Given that my question is: what happens to time variable? Do authors predict averaged velocity or the velocity at some particular moment in time  or something else entirely?*
>
> Exactly, all spatiotemporal fields are averaged over time. We added this detail to Appendix B to make this clearer

---

> ### Author Response · Authors · 2025-09-03
> **Rebuttal to review of Paper5252 by Reviewer b9ft (3/4)**
>
> ## Claim on vorticity
> - *It seems to me that the claim on vorticity is a somewhat overemphasised. In equation (11) on page 12 authors explain that they simply approximate vorticity by postprocessing velocity with finite difference approximation.
> I have several objections on the significance of this contribution:*
>     - 1. *Vorticity linearly depend on velocity, so to recover the vorticity is a matter of postprocessing.*
>     - 2. *The procedure itself is straightforward and not novel. Physics-informed neural network are available that enforce the divergence-free condition exactly, e.g., https://arxiv.org/abs/2210.01741*
>
>     *I suspect that the finite-difference approximation alone with no end-to-end training will not provide good results for vorticity prediction from velocity. However, I believe a tiny GNN can be trained on predicted velocities to recover good approximation to vorticity. Did the authors try this kind of postprocessing by training additional simple model?*
>
>     *Can the authors please comment on why they consider this part of their work as important contribution?*
>
>  For a justification of why we consider the divergence-free vorticity as an important contribution, please see our response to reviewer eEaY. Note that standard transformers map sets to sets and are therefore not neural fields. Our formulation as a conditional neural field via anchor attention enables imposing physics constraints, which we consider very important, and we view the divergence-free vorticity as an example of such constraints.
>
> Regarding the reviewer's objection, "Vorticity linearly depends on velocity, so to recover the vorticity is a matter of postprocessing.":
> It is correct that, in principle, vorticity could be derived from the velocity as a simple postprocessing, as vorticity is defined via the curl of the velocity field. But the resulting vorticity field would have enormous errors compared to the ground truth. We found it essential to train the derived vorticity together with all other fields rather than merely postprocessing it. We used a two-stage setup with pretraining (no vorticity) and finetuning (including vorticity), because training the derived vorticity requires 32-bit precision and is therefore slower.
>
> Note also that the reason why omitting fine-tuning would lead to enormous errors is not due to the finite-difference approximation, but because neural networks don't fit the derivatives (unless constrained or trained to do so).
> We found that the finite-difference approximation is indeed quite accurate (using 32-bit precision).
>
> The reviewer correctly pointed out that Richter-Powell et al. already introduced a method to enforce the divergence-free condition (via a generalization of the curl to higher dimensions). We therefore weakened our claims on the novelty of this part of our work. Note, however, that this work is already included in our related work section, and we acknowledge that it is very closely related.
>
> We used a finite-difference approximation rather than the exact Jacobian computation for two reasons: 1) Jacobian computation via automatic differentiation and gradient-based optimization thereof is not possible with FlashAttention since the required derivatives have not been implemented for these custom kernels; 2) the Pytorch compiler unfortunately does not manage to compile efficient kernels for computing the independent 3x3 Jacobians per each query point, resulting in out-of-memory even at small scales. We verified that the XLA compiler in Jax does not have this problem, but our experiment framework is based on PyTorch.
>
> Regarding the question about training a smaller model on top of the predictions of AB-UPT. We have not tried this and this sounds interesting, but we do not see the benefit this approach would bring. As mentioned above, the finite-difference approximation is not the bottleneck, and the learned velocity must be trained jointly with the vorticity in order to yield both accurate field values and field derivatives (for the curl).

---

> ### Author Response · Authors · 2025-09-03
> **Rebuttal to review of Paper5252 by Reviewer b9ft (4/4)**
>
> ## Claim on CAD geometry predictions
> - - *The mesh is required to obtain a data from numerical simulation. Is it the case that at best the mesh can be ignored only at inference? if this is true, I suggest to specify this explicitly in the paper.*
>
> It is true that meshing is still required to obtain training data, which we explicitly state as "The model is then trained by using surface/volume positions from the CFD mesh as query points on which a loss is calculated," in Section 4.6. We added a clarification that the mesh is optional only at inference time to the abstract and Section 4 introduction, which previously could suggest that the mesh is also optional
> for training.
> - - *The claim that CAD data can be used seems to be problem-specific. Suppose I represent some terrain (mountains and valleys) with CAD file and is interested in wind flow in this environment. Alternatively, one may consider a wind flow in the city layout. The selection of points on the surface is straightforward and can be done as authors suggest. The selection of points in the volume does not seem to be straightforward and the proper selection resembles an initial stages of mesh construction. How the authors suggest to select points in the volume when geometry is more involved?*
>
> We find that sampling volume anchors from a simple regular grid already obtains quite good results (as shown in Section 4.6), this means that volume points might also be located within the car. However, this is not a problem, as the model does not need to learn the solution at anchor points, but only at query points, which would always be outside the car (or above ground level in the terrain example) when training the model. This is conceptually similar to register tokens used in computer vision architectures ("Vision Transformers Need Registers", Darcet et al., ICLR 2024) where tokens from non-informative regions in an image store information for other regions.
>
> As the geometry in DrivAerML can also be quite involved with small details (e.g., underbody of the car or tires), we believe that sampling volume anchors from a regular grid would also produce sensible results on involved terrains to study wind flow. If a cheap/coarse-resolution meshing algorithm is available, volume anchors could also be sampled from that, which would most likely be better than the regular grid sampling but worse than sampling from the CFD mesh.
> ## Scaling
>
> - *In my view we see (Figure 8) quick saturation of performance measure when the size of the architecture or the size of the dataset are increased. What is the opinion of the authors on that? Do they expect that it will be possible to increase accuracy significantly (on a few orders) by increasing the size of the transformer and the size of the dataset?*
>
> Scaling laws of transformers are commonly described via a power-law relationship (e.g., "Training Compute-Optimal Large Language Models", Hoffmann et al.). The scaling curves in Figure 8 follow a similar pattern. Notably, the final models trained according to these scaling laws scale multiple axes, such as model size, training duration, and dataset size, where we show single-axis scaling in Figure 8, as we do not have access to, e.g., more data, which would be necessary to avoid overfitting when training larger models (e.g., model dim 768). As an alternative, we show how scaling multiple axes of our model at once performs and observe a similar pattern as in large-scale language models, where scaling multiple axis at once obtains the best performance.
>
> To better illustrate the impact of scaling, we added a table with the best performing models in each individual axis and model scaled on multiple axes in a table with corresponding performance metrics in addition to the test loss. This shows that scaling does significantly improve performance metrics, most notably it improves the DrivAerML vorticity relative L2 error from 35.1 to 18.8\%. We would expect further accuracy gains from further scaling, particularly from scaling the dataset size. However, as this is very expensive, we do not further explore this direction.
>
> |                | #Tokens | Model dim | Model depth | Test loss | surface pressure  | velocity    | vorticity     |
> |----------------|---------|-----------|-------------|-----------|-------|-------|-------|
> | **AB-UPT**     | 16K     | 192       | 12          | 0.1314    | 3.82  | 5.93  | 35.1  |
> |                |         |           |             |           |       |       |       |
> | **AB-UPT**     | **128K**| 192       | 12          | 0.1300    | 3.61  | 5.69  | 32.7  |
> | **AB-UPT**     | 16K     | **384**   | 12          | 0.1214    | 3.55  | 5.20  | 25.4  |
> | **AB-UPT**     | 16K     | 192       | **60**      | 0.1212    | 3.47  | 5.20  | 21.0  |
> |                |         |           |             |           |       |       |       |
> | **AB-UPT**     | **32K** | **384**   | **24**      | **0.1169**| 3.38  | 4.87  | 18.8  |

---

> > ### Comment · Reviewer_b9ft · 2025-09-19
> >
> > I would like to thank the authors for their detailed reply. All of my questions are sufficiently addressed. I still have some disagreement with the authors viewpoint (e.g., I believe there is no “scaling law” according to the results of the authors), but overall provided clarifications made the contribution more clear to me and, hopefully, for other readers alike. My recommendation is to accept the paper.

---

### Review · Reviewer_eEaY · 2025-08-12

**Summary Of Contributions:**

The paper introduces AB-UPT, a Transformer-based model for CFD surrogate modelling. Combining several architectural improvements with engineering tweaks, the proposed model shows strong empirical performance on several large-scale CFD benchmark problems. The model features scaling laws and can do inference on CAD geometries without the need of tailored meshing.

**Additional Comments:**

Will authors release the code of the project to public, or maybe a prototype implementation of the model architecture?

**Audience:**

Yes

**Audience Explanation:**

The findings will interest TMLR readers in ML for scientific computing, and efficient Transformer design in scientific problems.

**Claims And Evidence:**

Yes

**Claims Explanation:**

Overall, most of the claims are backed by clear evidence.

Strength:

* The manuscript is well-written, and the ablations are thorough; key design choices are well-motivated and empirically validated.

* Efficient architecture (pooling + anchor-based attention) enables scalable training with decent accuracy. The authors have also introduced techniques like FlashAttention / mixed predicision training /chunked inference into model training and inference, while these techniques are pretty common in computer vision and LLM models, these techniques are underused in neural surrogate community and this paper offers an empirical reference for adopting them.

* The proposed model delivers strong results, outperforming state-of-the-art baselines on multiple datasets.

Weakness:

* Several core components—e.g., Perceiver-style cross-attention and GNO-style aggregation are well established. The contribution is more in integration and scaling rather than new primitives.

* The model architecture is tailored for CFD problem specifically. The parameterization requires six model forward to estimate vorticity.

**Requested Changes:**

1. For the mixed precision training part, ignoring the compute cost, what will be the accuracy of the model if adopting a float32 precision?

2. In terms of pointwise metrics like L2 error, it seems that parameterizing the model with divergence-free prediction does not necessarily yield significant improvement, can the authors elaborate more on benefits here?

3. Conceptually, Transovler and AB-UPT use similar attention paradigms (standard softmax attention), where AB-UPT uses GNO aggregation to sparsify nodes and Transolver uses learnable slicing to sparsify. Why is it categorized as linear attention variant?

---

> ### Author Response · Authors · 2025-09-03
> **Rebuttal to review of Paper5252 by Reviewer eEaY (1/2)**
>
> We want to thank the reviewer for their thoughtful review. We tried to address all the requested changes and discuss all weaknesses. We will now discuss each point individually.
>
> ## Weaknesses
>
> - *Several core components—e.g., Perceiver-style cross-attention and GNO-style aggregation are well established. The contribution is more in integration and scaling rather than new primitives.*
>
> We agree with the reviewer that many components of AB-UPT are well established, and we also see integrating these tried-and-tested components into a fast and accurate architecture as our core contribution.
> However, we would also argue that the formulation of a joint self- and cross-attention mechanism as anchor attention is a novel contribution.
> We are not aware of existing components that enable omitting the cross-attention component of anchor-attention during training and only using it for inference (see e.g., Appendix A.1).
>
> - *The model architecture is tailored for CFD problem specifically. The parameterization requires six model forward to estimate vorticity.*
>
> We use CFD datasets to develop our model architecture, but we would argue that the AB-UPT model (and its components) is quite general and easy to extend to different tasks.
> For example, by removing either the surface/volume branches, one could easily train the model on "unimodal" simulations where it is not straightforward to split the data into both surface and volume.
> This model would be equivalent to Row 5 in Table 3 ("+ decouple geometry encoding"), which obtains roughly on-par results with AB-UPT at the cost of more runtime.
>
> To estimate vorticity via the divergence-free formulation, it is indeed necessary to do six additional forward passes for the volume positions.
> However, these additional forward passes can be computed in parallel (batched) and the corresponding tokens only use cross-attention to the anchor tokens.
> Note also that, by default, when divergence-free vorticity is not required, we predict the vorticity directly, which requires only one forward pass.
>
> ## Request changes
>
> - *For the mixed precision training part, ignoring the compute cost, what will be the accuracy of the model if adopting a float32 precision?*
>
> Our model design and implementation achieve high accuracy with float16 mixed-precision training. float32 precision achieves the same accuracy, aside from minor random variance at vastly larger runtimes.**
>
> | Method              | Surface pressure | Velocity | Vorticity | Wallshearstress | Volume pressure | Training time |
> | ------------------- | ---------- | ------- | ----------------------- | --------------------- | ---------- | ------------- |
> | AB-UPT (full fp32)  | 3.78       | 5.94    | 36.1                    | 7.25                  | 6.07       | 33.6h         |
> | AB-UPT (mixed-fp16) | 3.82       | 5.93    | 35.1                    | 7.29                  | 6.08       | **6.9h**      |
>
> We added this experiment as Appendix A.12 together with a short summary in the introduction of Section 4.
>
> - *In terms of pointwise metrics like L2 error, it seems that parameterizing the model with divergence-free prediction does not necessarily yield significant improvement, can the authors elaborate more on benefits here?*
>
> The divergence-free vorticity formulation offers a physics-adherent way to predict the vorticity without losing predictive accuracy. It shows how formulating models as conditional neural fields (via anchor attention) enables imposing physics constraints onto the model by estimating differential operators via finite differences (or automatic differentiation). We show this capability on a comparatively simple use case of predicting divergence-free vorticity.
> Note, however, that our neural field formulation can be leveraged to impose additional physics constraints, such as mass conservation of the velocity field.
>   That being said, imposing mass conservation comes with additional challenges that we leave to future work: while a mass-conserving velocity field is divergence-free by definition, the Helmholtz-Hodge Decomposition tells us that the velocity field may have an additive component that is simultaneously divergence-free and vorticity-free (i.e,. harmonic), which can not be represented exclusively via the curl operator. This is in contrast to the vorticity field, which does not have a harmonic component, since it is literally defined as the curl of the velocity field. We added a discussion to this point in Section 5 ("Physics constraints via neural field formulation").

---

> ### Author Response · Authors · 2025-09-03
> **Rebuttal to review of Paper5252 by Reviewer eEaY (2/2)**
>
> - *Conceptually, Transolver and AB-UPT use similar attention paradigms (standard softmax attention), where AB-UPT uses GNO aggregation to sparsify nodes and Transolver uses learnable slicing to sparsify. Why is it categorized as linear attention variant?*
>
> While Transolver is similar to AB-UPT in the fact that it uses standard softmax attention in sliced space, we argue that Transolver is conceptually more similar to linear attention mechanisms like the ones used in, e.g., OFormer, as it sacrifices model capacity for the benefit of linear runtime (assuming the number of slices is much lower than the number of tokens).
>
> ## Additional Comments
> - *Will authors release the code of the project to public, or maybe a prototype implementation of the model architecture?*
>
> Thanks for the suggestion! The inference code (including model architecture, model weights, pre-processing, dataloading, and metric calculation) is already part of the submission and available here: [https://anonymous.4open.science/r/ABUPT-tutorial-A6DC/README.md](https://anonymous.4open.science/r/ABUPT-tutorial-A6DC/README.md). Upon acceptance, we will make the code publicly available. It was provided as a footnote in Section 4. We moved it to the main text.

---

> > ### Comment · Reviewer_eEaY · 2025-09-22
> >
> > I appreciate the authors’ detailed response and clarification. My questions have been addressed. I think the paper could be a worthy addition to the community.

---

> > > ### Comment · Action_Editor_GSYv · 2025-09-22
> > >
> > > Dear reviewer,
> > >
> > > Thank you for your engagement in the reviewing process. For the procedure to go through, could you formally submit a "Recommendation"?
> > >
> > > Thank you!
> > > The AE

---

### Review · Reviewer_RpMp · 2025-08-26

**Summary Of Contributions:**

This paper introduces AB-UPT (Anchored-Branched Universal Physics Transformers), a transformer-based neural surrogate architecture for large-scale computational fluid dynamics (CFD) in automotive aerodynamics. The contributions are threefold:
- Multi-branch design that decouples geometry encoding, surface prediction, and volume prediction.
- Anchor attention mechanism, reducing quadratic attention cost to linear in the number of mesh queries, enabling scaling to meshes with up to 150M cells on a single GPU.
- Physics consistency via architectural constraints, enforcing divergence-free vorticity by construction.

### Strengths:

- Strong architectural innovations addressing scalability, efficiency, and physics constraints.
- Thorough experimental evaluation across multiple datasets, with state-of-the-art accuracy and practical runtime gains.
- Demonstrates real engineering relevance (drag/lift predictions, meshless CAD-to-flow capability).

### Weaknesses:

- Generalization beyond training distribution is untested. All datasets are similar in nature (very similar designs). It remains unclear whether AB-UPT would extrapolate to flows with different Reynolds, boundary conditions, or completely different geometries (e.g., OOD designs).
- Reliance on expensive high-fidelity data. Training still requires hundreds of HRLES simulations, which cost millions of CPU hours. Without such data, adoption in domains with scarcer resources may be limited.
- Potential fragility of anchor sampling. The random anchor selection strategy may not optimally capture rare but important local phenomena (like vortices or flow separations), raising concerns about robustness to distribution shifts or underrepresented cases!
- Scaling bottleneck remains with anchors. While reducing quadratic cost, the approach still scales quadratically in the number of anchors, which may become limiting if much finer fidelity or broader domains are required.

**Additional Comments:**

- While divergence-free enforcement is an elegant addition, future work could extend such hard constraints to other invariants (e.g., wall boundary conditions, energy conservation).

- The CAD-only experiment is promising, but results show non-negligible degradation. A brief discussion of how to mitigate this (e.g., hybrid approaches using lower-fidelity meshes) would be useful.

**Audience:**

Yes

**Audience Explanation:**

The paper introduces novel architectural concepts (anchor attention, multi-branch physics transformers, divergence-free enforcement) that advance the field of neural surrogate modeling. These findings are relevant to both the machine learning community (for methods innovation) and scientific computing practitioners (for practical CFD surrogates). However, interest might be more concentrated among researchers focused on CFD and physics-informed ML, given that broader generalization claims remain to be tested.

**Broader Impact Concerns:**

No major ethical risks identified. The main broader impact relates to improving aerodynamic design efficiency. Potential positive impacts include reduced environmental costs through more efficient vehicles. A limitation worth noting is that reliance on large high-fidelity CFD datasets means access may remain restricted to well-funded labs/companies, raising equity concerns in who can reproduce or extend this line of research.

**Claims And Evidence:**

Yes

**Claims Explanation:**

The authors convincingly demonstrate state-of-the-art results on three datasets, provide ablation studies that justify architectural choices, and evaluate both accuracy and runtime efficiency. The claims about divergence-free enforcement and mesh-independence are well supported. However, some stronger claims about generality and real-world deployment are less substantiated. For example, there is no evidence of performance under out-of-distribution conditions (e.g., different Reynolds numbers or very different geometries). While this does not undercut the main contributions, it should temper the reader’s expectations about immediate general-purpose applicability.

**Requested Changes:**

### Critical:

- Please include comparisons with Transolver++ and TripNet for the volume velocity field prediction task. These are important recent baselines in large-scale CFD surrogate modeling, and without these comparisons it is hard to fully judge the advantages of AB-UPT on volume quantities.
- It would be very helpful to visualize predictions versus ground-truth CFD using 2D slices of the 3D flow fields (e.g., velocity magnitude or individual components on a cross-sectional plane through the wake or near-body region). Numerical error metrics are comprehensive, but qualitative plots are standard in CFD to demonstrate whether small-scale flow structures (e.g., recirculation, vortices) are preserved. Including such comparisons would greatly strengthen the presentation and allow readers to assess the physical fidelity of AB-UPT’s predictions.
- Add discussion on limitations of generalization: how does AB-UPT behave when test conditions differ from training (geometry classes, Reynolds number, etc.)? Explicitly stating this limitation would make the scope of claims clearer.
- Acknowledge a limitation of the DrivAerML dataset: while it is high-fidelity, all geometries are relatively similar (variants of one base car) and are simulated under the same boundary conditions. This means AB-UPT’s excellent performance may partly reflect the dataset’s homogeneity, and future work should test on more diverse geometries and flow conditions.


### Would strengthen the work:

- Discuss strategies for smarter anchor selection (e.g., physics-guided, uncertainty-based) to mitigate risks of missing critical regions.
- Add forward-looking commentary on robustness to distribution shifts and what mechanisms (data augmentation, multi-fidelity training, or uncertainty quantification) might help.
- Address the scaling bottleneck with anchor count and mention possible extensions (e.g., hierarchical anchors, linear attention mechanisms).

---

> ### Author Response · Authors · 2025-09-03
> **Rebuttal to review of Paper5252 by Reviewer RpMp (1/3)**
>
> We want to thank the reviewer for their thoughtful review. We tried to address all the requested changes. We will now discuss each mention point individually.
>
> ## Weaknesses
>
> - *Generalization beyond training distribution is untested. All datasets are similar in nature (very similar designs). It remains unclear whether AB-UPT would extrapolate to flows with different Reynolds, boundary conditions, or completely different geometries (e.g., OOD designs).*
>
> Testing AB-UPT on generalizations to different Reynolds numbers, boundary conditions, or completely different geometries is a very intriguing direction. However, the sheer costs of generating high-fidelity automotive CFD simulations tend to limit the variability in the dataset. Datasets that offer higher diversity typically sacrifice simulation fidelity. For example, the DrivAerNet++ dataset offers different geometries and car types at reduced fidelity. In our work, we focused on modeling high-fidelity HRLES simulations, where we are not aware of any dataset that offers more variability at similar fidelity. Nevertheless, we find this an important aspect and, therefore, added additional results for DrivAerNet++ in Appendix A.4, where AB-UPT also shows strong performances.
>
>
> - *Reliance on expensive high-fidelity data. Training still requires hundreds of HRLES simulations, which cost millions of CPU hours. Without such data, adoption in domains with scarcer resources may be limited.*
>
> We totally agree with the reviewer that reliance on high-fidelity data is a limitation. However, this limitation is a general limitation of data-driven neural surrogate models and orthogonal to the architecture and modeling paradigms of AB-UPT. We provide a discussion related to this in Section 5, where we discuss how improvements to the CFD simulation, such as Lattice Boltzmann Method approaches, could help alleviate this limitation of machine learning-based surrogate models.
>
> - *Potential fragility of anchor sampling. The random anchor selection strategy may not optimally capture rare but important local phenomena (like vortices or flow separations), raising concerns about robustness to distribution shifts or underrepresented cases!*
>
> While investigating robustness to distribution shifts or underrepresented cases would be interesting, it is quite difficult with the currently available datasets, given that the geometric variability is relatively limited and generating new data would be really expensive. To investigate the fragility of anchor sampling, we choose a fixed set of query points and randomly sample anchors, allowing us to quantify the variance of the predictive accuracy with differently chosen anchor points. We provide this new study in Appendix A.8, which shows that our model is relatively robust against the anchor sampling procedure.
>
>
> - *Scaling bottleneck remains with anchors. While reducing quadratic cost, the approach still scales quadratically in the number of anchors, which may become limiting if much finer fidelity or broader domains are required.*
>
> This is a very valid point, which we discuss in Section 5 of the paper. Computational costs of anchor attention still increase quadratically with the number of anchors. If an extremely high fidelity or broader domains require a large number of anchors (e.g., 1 million), the computational costs would become extremely large. We found that even for CFD simulations using 150M mesh cells, they can be captured fairly well with a comparatively small number of anchor points (e.g., 16K) where AB-UPT shows a favorable cost vs accuracy tradeoff compared to other models where investing more compute into training other models (e.g., by increasing the number of points to 32K) does not outperform AB-UPT with similar or lower costs. This suggests that AB-UPT can capture more information from the low-resolution input than other approaches.

---

> > ### Comment · Reviewer_RpMp · 2025-09-20
> > **Reviewer Comment on Rebuttal – Paper5252**
> >
> > Thank you for the detailed rebuttal and for addressing my concerns. I appreciate the additional experiments on DrivAerNet++, and I found those results particularly interesting in demonstrating AB-UPT’s robustness beyond the original datasets. The discussion around anchor sampling robustness and scaling trade-offs was also helpful.
> >
> > Overall, the authors have satisfactorily addressed the main issues I raised. I recommend the paper for acceptance and thank the authors for both their contributions and the thoughtful rebuttal.

---

> ### Author Response · Authors · 2025-09-03
> **Rebuttal to review of Paper5252 by Reviewer RpMp (2/3)**
>
> ## Requested Changes
>
> - *Please include comparisons with Transolver++ and TripNet for the volume velocity field prediction task. These are important recent baselines in large-scale CFD surrogate modeling, and without these comparisons it is hard to fully judge the advantages of AB-UPT on volume quantities.*
>
> We did not consider Transolver++ as a baseline as it did not have a publicly available implementation at the time of writing, and our reproduction attempts resulted in Transolver++ performing worse than Transolver (and consequently performing much worse than AB-UPT). We discussed this shortly in the main paper and provided experiments in Appendix C.2. As a public implementation of Transolver++ was released recently, we additionally ran Transolver++ in our benchmark setting, where we observe similar behavior as with our own reimplementation, namely, consistently worse performance than Transolver. Consequently, we use the vanilla Transolver as a baseline in the main paper.
>
> To further evaluate AB-UPT on a dataset with more geometric variability, we added experiments on DrivAerNet++ in Appendix A.4. We also compare our DrivAerNet++ results to the ones reported in TripNet, where AB-UPT outperforms TripNet (see Appendix A.5).
>
> - *It would be very helpful to visualize predictions versus ground-truth CFD using 2D slices of the 3D flow fields (e.g., velocity magnitude or individual components on a cross-sectional plane through the wake or near-body region). Numerical error metrics are comprehensive, but qualitative plots are standard in CFD to demonstrate whether small-scale flow structures (e.g., recirculation, vortices) are preserved. Including such comparisons would greatly strengthen the presentation and allow readers to assess the physical fidelity of AB-UPT’s predictions.*
>
> Thanks for the suggestion! We decided to go with a visualization of the velocity streamlines instead of 2D slices of the 3D flow field.
> After some experimentation, we concluded that 2D visualizations weren't effective at conveying the message we wanted to get across. We've included two plots in Figure 12 of Appendix A.13 that show the velocity streamlines: one using the ground truth velocity and another using the velocity predicted by AB-UPT. It's clear from these plots that the two are nearly identical.
>
> - *Add discussion on limitations of generalization: how does AB-UPT behave when test conditions differ from training (geometry classes, Reynolds number, etc.)? Explicitly stating this limitation would make the scope of claims clearer.*
>
> We agree that further evaluation of more generalization capabilities would be an interesting study. However, the limited availability of current datasets makes this difficult, as generating high-fidelity datasets with many variabilities is expensive, so many datasets trade off fidelity with variability to make it computationally feasible. We added this as a discussion point to Section 5. Additionally, we evaluate AB-UPT on DrivAerNet++, which uses low-fidelity RANS simulations but has more geometric diversity than DrivAerML. The results in Appendix A.4 show that AB-UPT also performs well on this dataset.
>
> - *Acknowledge a limitation of the DrivAerML dataset: while it is high-fidelity, all geometries are relatively similar (variants of one base car) and are simulated under the same boundary conditions. This means AB-UPT’s excellent performance may partly reflect the dataset’s homogeneity, and future work should test on more diverse geometries and flow conditions.*
>
> We added a paragraph to Section 5 to discuss this limitation.
>
> - *Discuss strategies for smarter anchor selection (e.g., physics-guided, uncertainty-based) to mitigate risks of missing critical regions.*
>
> More sophisticated anchor selection processes are definitely an interesting future direction, including physics-guided (e.g., dense sampling at boundary regions), uncertainty-based, or even learn it end-to-end. We added a paragraph to address this in Section 5 ("Further improvements of AB-UPT and neural surrogates")
>
> - *Add forward-looking commentary on robustness to distribution shifts and what mechanisms (data augmentation, multi-fidelity training, or uncertainty quantification) might help.*
>
> We believe all of the mentioned techniques could improve performance further, but are not directly related to AB-UPT, rather apply to all kinds of neural surrogate models. Therefore, we consider these to be future work. We added a paragraph to address this in Section 5 ("Further improvements of AB-UPT and neural surrogates").

---

> ### Author Response · Authors · 2025-09-03
> **Rebuttal to review of Paper5252 by Reviewer RpMp (3/3)**
>
> - *Address the scaling bottleneck with anchor count and mention possible extensions (e.g., hierarchical anchors, linear attention mechanisms).*
>
> We would like to draw attention to the "The bottleneck of quadratic complexity of self-attention for anchor tokens" paragraph in Section 5, where we discuss the limitation of quadratic complexity of anchor tokens in detail. We also suggest ball-tree attention (Zhdanov, ICML 2025) as an extension of AB-UPT there, where ball-tree attention has linear complexity by limiting the interaction radius to a local ball and can be made hierarchical by combining multiple balls together.
>
> (Zhdanov, ICML 2025): https://icml.cc/virtual/2025/poster/45529
>
> ## Additional Comments
>
> - *While divergence-free enforcement is an elegant addition, future work could extend such hard constraints to other invariants (e.g., wall boundary conditions, energy conservation).*
>
> We also believe that enabling differential operators via the neural field formulation (via finite-differences estimation or direct calculation via automatic differentiation) has potential beyond divergence-free vorticity predictions. However, formulating physics constraints can be quite challenging on its own, which is why we leave it for future work. We added a discussion to this point in Section 5 ("Physics constraints via neural field formulation").
>
> - *The CAD-only experiment is promising, but results show non-negligible degradation. A brief discussion of how to mitigate this (e.g., hybrid approaches using lower-fidelity meshes) would be useful.*
>
> Thanks for pointing this out. This is a very interesting future direction. We expanded the discussion in Section 4.6 to incorporate this important point. Additionally, we found an error in the evaluation pipeline of the CAD-only experiment, resulting in a significantly worse vorticity L2 error for DrivAerML. Although it does not change that the results are worse, the drop in DrivAerML vorticity L2 error is not as severe (fixing the error decreased the error from 76.5\% to 51.2\% whereas using the CFD mesh obtains 35.2\%).

---

### Author Response · Authors · 2025-09-03
**Rebuttal to reviews and paper revision**

Dear reviewers and Action Editor,

Thank you for your thoughtful reviews and feedback. We have considered all of the concerns that were raised and have updated the paper accordingly. We have replied to each review individually, providing detailed answers to address your concerns.

All text that has been added or updated since our last submission is highlighted in green. Text that has been removed is indicated with a red strikethrough.

We also want to highlight two general updates we've made to the paper:
- We have included the DrivAerNet++ dataset as an additional benchmark in Appendix A.4.
- We have corrected a mistake in Figure 2, regarding the mesh cell sizes for the benchmark dataset.

We look forward to the discussion.

Thanks,
The authors

---

> ### Comment · Action_Editor_GSYv · 2025-09-10
>
> Dear reviewers,
>
> Please consider the answers and revisions provided below by the authors. Based on these elements, and optionally on further discussion you would want with the authors, please formulate your recommendation.
>
> Kind regards,
> The AC

---

### Decision · Action_Editor_GSYv · 2025-09-26

**Recommendation:** Accept as is

**Audience:**

Yes

**Audience Explanation:**

This paper is among the first to scale neural CFD surrogates to operational scale. More broadly, the novel architecture proposed in the paper could certainly be interest to the broader community interested in scientific simulations.

**Claims And Evidence:**

Yes

**Claims Explanation:**

The claims of the manuscript are supported by empirical evidence. Further results were provided during the discussion to the satisfaction of the reviewers.

---

> ### Author Response · Authors · 2025-10-10
> **Camera-ready revision**
>
> Dear AC,
> We uploaded the camera-ready version of our paper. Thanks again to all reviewers and the AC for their effort in reviewing our paper!